# A unifying Bayesian framework for merging X-ray diffraction data

**Kevin M. Dalton** [1], **Jack B. Greisman** [1] **& Doeke R. Hekstra** [1,2] ✉

Novel X-ray methods are transforming the study of the functional dynamics of biomolecules. Key to this revolution is detection of often subtle conformational changes from diffraction data. Diffraction data contain patterns of bright spots known as reflections. To compute the electron density of a molecule, the intensity of each reflection must be estimated, and redundant observations reduced to consensus intensities. Systematic effects, however, lead to the measurement of equivalent reflections on different scales, corrupting observation of changes in electron density. Here, we present a modern Bayesian solution to this problem, which uses deep learning and variational inference to simultaneously rescale and merge reflection observations. We successfully apply this method to monochromatic and polychromatic single-crystal diffraction data, as well as serial femtosecond crystallography data. We find that this approach is applicable to the analysis of many types of diffraction experiments, while accurately and sensitively detecting subtle dynamics and anomalous scattering.

X-ray crystallography has revolutionized our understanding of the molecular basis of life by providing atomic-resolution experimental access to the structure and dynamics of macromolecules and their assemblies. In an X-ray diffraction experiment, the electrons of a molecular crystal scatter X-rays, yielding patterns of constructive interference recorded on an X-ray detector. The resulting images contain discrete spots, known as reflections, with intensities proportional to the squared amplitudes of the Fourier components (structure factors) of the crystal's electron density. Each structure factor reports on the electron density at a specific spatial frequency and direction, indexed by triplets of integers termed Miller indices. Estimates of the amplitudes and phases of these structure factors allow one to reconstruct the 3D electron density in the crystal by Fourier synthesis.

Based on these principles, advances in X-ray diffraction now permit direct visualization of macromolecules in action[1] using short X-ray pulses generated at synchrotrons[2,3] and X-ray Free-Electron Lasers (XFELs)[4,5]. The full realization of the promise of these methods hinges on the ability to separate signals in X-ray diffraction that result from subtle structural changes from a multitude of systematic errors that can be specific to a crystal, X-ray source, detector, or sample environment[6]. Even under well-controlled experimental conditions,

redundant reflections are expressed on the X-ray detector with different scales (Fig. 1). These scales depend non-linearly on the context of each observed reflection as illustrated in Fig. 1b-d. For example, beam properties like intensity fluctuations[7] and polarization[8], crystal imperfections like mosaicity[9] and radiation damage[10], and absorption and scattering of X-rays by material around the crystal all modulate the measured diffraction intensities in a manner, which varies throughout the experiment.

Traditionally, these artifacts are accounted for by estimating a series of scale parameters that are intended to explicitly model the physics of the sources of error[6,11–13] (see the Supplementary Note for a description of crystallographic data reduction). The observed intensities are then corrected by each scale parameter to yield scaled intensities. To obtain consensus merged intensities, equivalent observations are then merged by weighted averaging assuming normally distributed errors. This approach thus uses a series of simplifying 'data reduction' steps that work well for standard diffraction experiments at synchrotron beamlines, but are less suited for a rapidly evolving array of next-generation X-ray diffraction experiments.

Although intensities are proportional to squared structure factor amplitudes, negative intensities can result from processing

[1]Department of Molecular & Cellular Biology, Harvard University, Cambridge, MA 02138, USA. [2]John A. Paulson School of Engineering and Applied Sciences, Harvard University, Cambridge, MA 02138, USA. ✉e-mail: doeke_hekstra@harvard.edu

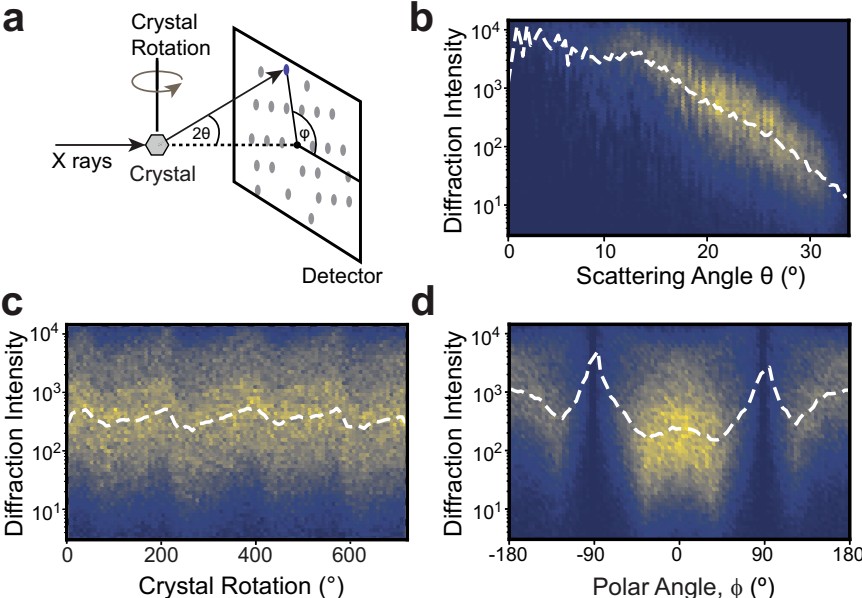

**Fig. 1 | The intensity of observed reflections depends on physical factors.**
**a** Geometry of a conventional diffraction experiment: a crystal (shown as a hexagon) scatters an incident X-ray beam and yields a pattern of reflections (gray spots) on a detector. Three metadata of the measurements are indicated: scattering angle $2\theta$, crystal rotation angle, and polar angle, $\phi$. **b-d** depict the dependence of the observed intensity distributions on these metadata for a hen egg-white lysozyme dataset. The 2-dimensional histograms show the number of counts in each of bins on a logarithmic scale. White dashed lines indicate the median intensity in each $x$-axis bin. Reflections with $I/\sigma_I < = 0$ were discarded from this analysis. **b** Diffraction intensities decrease with increasing scattering angle, or resolution. **c** Diffraction intensities vary with crystal rotation angle, a proxy for cumulative radiation dose and variations in diffracting volume. **d** Diffraction intensities depend on polar angle due to polarization of the X-ray source and absorption effects.

steps such as background subtraction and may persist through scaling and merging. French-Wilson corrections are commonly applied to ensure that inferred structure factor amplitudes are strictly positive[14]. Rather than being based on a physical model of diffraction, this step is based on a Bayesian argument. Namely, structure factor amplitudes are positive and can be expected to follow the so-called Wilson distribution[15]. In a Bayesian sense, the Wilson distribution serves as a prior probability distribution, or prior. This prior can be combined with a statistical model of the true intensity given the observed merged intensity to obtain a posterior probability distribution, or posterior, of the true merged intensity that is strictly positive.

To address the needs of new X-ray diffraction experiments, here we introduce a Bayesian model which builds on the paradigm of French and Wilson[14]. We implement this model in an open-source software, Careless, which performs scaling, merging, and French-Wilson corrections in a single step by directly inferring structure factor amplitudes from unscaled, unmerged intensities. In our model, the probability calculation is "forward," predicting integrated intensities from structure factor amplitudes and experimental metadata. As a consequence, the analytical tractability of the inference is no longer a concern and the model relating structure factor amplitudes to integrated intensities can be arbitrarily complex and include both explicit physics and machine learning concepts. We demonstrate that this model can accurately and sensitively extract anomalous signal from single-crystal, monochromatic diffraction at a synchrotron, time-resolved signal from single-crystal, polychromatic diffraction at a synchrotron, and anomalous signal from a serial femtosecond experiment at an XFEL. Our analyses show that this single model can implicitly account for the physical parameters of diffraction experiments with performance competitive with domain-specific, state-of-the-art analysis methods. Although we focus on X-ray diffraction, we believe the same principles can be applied to any diffraction experiment.

## Results

### Accurate inference of scale parameters and structure factor amplitudes from noisy observations

In a typical diffraction experiment, reflection intensities are recorded along with error estimates and metadata, like crystal orientation, location on the detector, image number, and Miller indices. As shown in Fig. 1, the observed intensities vary systematically due to physical artifacts correlated with the metadata, causing the reflections to be related to the squared structure factor amplitudes by different multiplicative scale factors, or scales. These different scales must be accounted for in the analysis of diffraction experiments. Here, we present a probabilistic forward model of X-ray diffraction, which we implemented in a software package called Careless. As illustrated in Fig. 2a, this model can be abstractly expressed as a probabilistic graphical model. Specifically, the distribution of observable intensities, $I_{hi}$, for Miller index $h$ in image $i$, is predicted from structure factor amplitudes $F_h$ and scale factors $\Sigma$, which are estimated concurrently. We do so in a Bayesian sense—we estimate the posterior distributions of both the structure factor amplitudes and scale function. We approximate the structure factor amplitudes as statistically independent across Miller indices and use the Wilson distribution as a prior on their magnitudes[15].

By contrast, most contributions to scale factors vary slowly across the data set and are accounted for using a global parametrization. By default, this scale function, $\Sigma$, is implemented as a deep neural network, which takes the metadata as arguments and predicts the mean and standard deviation of the scale function for each observation (Fig. 2b). To describe measurement error, Careless supports both a normally distributed error model, and a robust Student's t-distributed error model. The implementation of Careless is described in further detail in the methods section. The full Bayesian model will typically contain tens of thousands of unique structure factor amplitudes and a dense neural network for the scale function. Use of Markov chain Monte Carlo methods[16], which sample from the posterior, would be computationally prohibitive. Instead, inference is made possible by

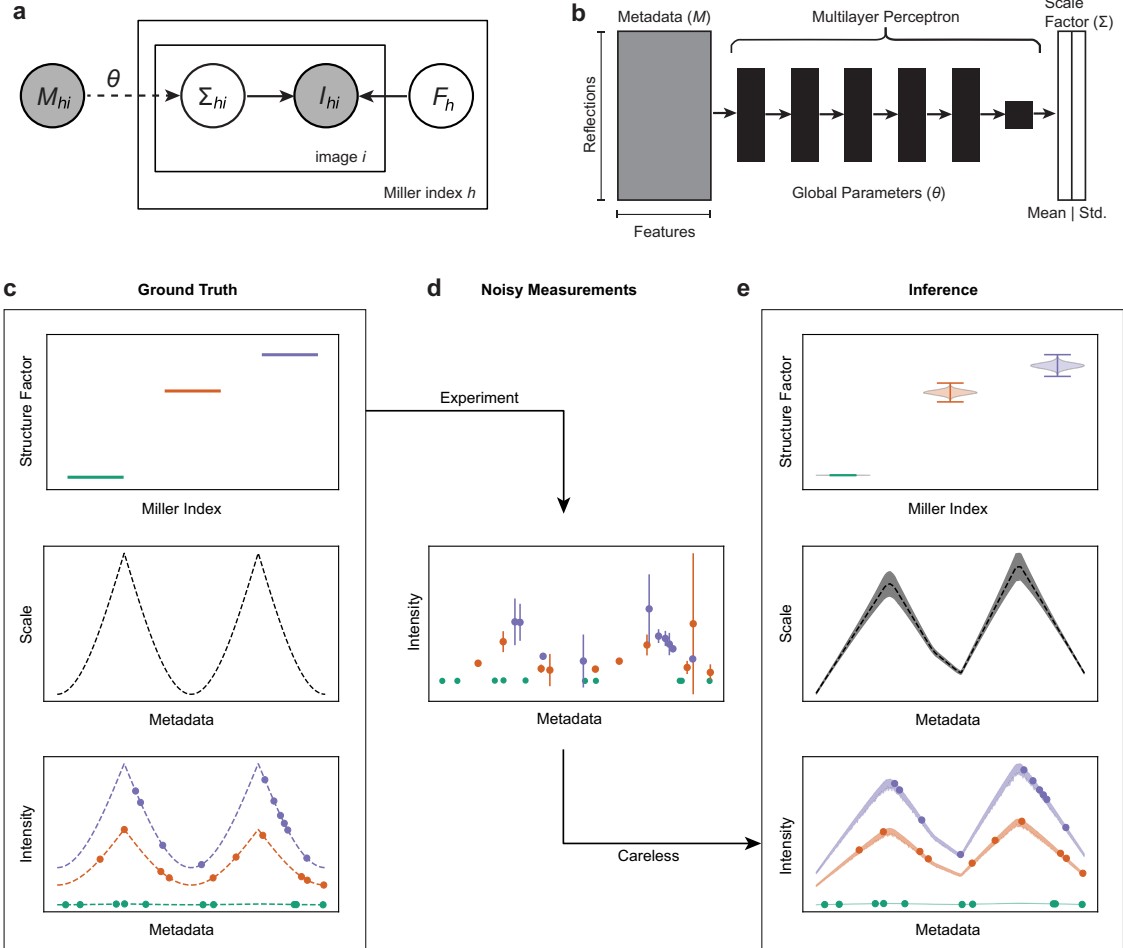

**Fig. 2 | Estimation of scales and structure factor amplitudes from simulated data. a** A probabilistic graphical model summarizes our basic statistical formalism: Careless calculates a probabilistic scale, Σ, as a function of the recorded metadata, $M$, and learned parameters, $\theta$. Observed intensities, $I_{h,i}$, for Miller index $h$ and image $i$ are modeled as the product of the scale and the square of the structure factor amplitude $F_h$, that is, as $\Sigma_{h,i} \cdot F_h^2$. **b** The global scale function that maps the recorded metadata to the probabilistic scale, Σ, takes the form of a multilayer perceptron. **c–e** Inference of scales and structure factors from simulated data comprising 10 draws from the ground truth model. (**c**) Input parameters for the simulated data were chosen to recapitulate the non-linear scales observed in diffraction data. **d** Noisy observations were generated from these input parameters, which reflects the measurement errors in a diffraction experiment (ten per structure factor). Shown are simulated measured values. Error bars indicate 95% confidence intervals. **e** This statistical model allows for joint, rather than sequential, inference of the posterior distributions of structure factor amplitudes and scales, and therefore of implied intensity (bottom panel). The violin plots in the top panel show the posterior probability with whiskers indicating the extrema of 10,000 samples drawn from the posterior distribution of the inferred $F$. Shaded bands indicate 95% confidence intervals around the posterior means. The posterior mean of the scale function is indicated as a dashed line (middle panel), and the posterior mean for the reflection intensities are shown as circles (bottom panel). For this toy example, the inferred values in (**e**) can be compared to the known ground truth in (**c**).

variational inference[17,18], in which the parameters of proposed posterior distributions are directly optimized.

We first illustrate the application of the Careless model using a small simulated dataset as shown in Fig. 2. As shown in Fig. 2c, we did so by generating true intensities for a toy "crystal" with 3 structure factors of different amplitudes in a mock diffraction experiment with a sharply varying scale function (similar to Fig. 1d). The observed intensities would be recorded with measurement error, yielding a small set of noisy observations (Fig. 2d). Using Careless, we can infer the posterior distributions of the structure factor amplitudes and of the scale factors, and therefore of the true intensities (Fig. 2e). The inferred parameters from Careless show a close correspondence with the true values used to simulate the noisy observations.

### Robust inference of anomalous signal from monochromatic diffraction

We next assessed the ability of Careless to extract small crystallographic signals from conventional monochromatic rotation series data (a detailed walk-through of each example is available at https://github.com/rs-station/careless-examples). To do so, we applied Careless to a sulfur single-wavelength anomalous diffraction (SAD) data set of lysozyme collected at ambient temperature[19]. It consists of a single 1,440 image rotation series collected in 0.5 degree increments at a low X-ray energy, 6.55 keV, at Advanced Photon Source beamline 24-ID-C. These data contain two sources of outliers (Fig. 3a). Most significantly, leakage from a higher energy undulator harmonic resulted in a second, smaller diffraction pattern in the center of each image. Additionally, a small number of reflections are located underneath a shadow from the beam stop mounting bracket near the edge of the detector (in the 2 to 2.2 Å range). These artifacts mean that there are many outliers at low resolution and some at high resolution in this data set. Conventional scaling and merging in Aimless or XDS[12,20] is successful for these data because these approaches use outlier rejection to explicitly identify and remove spurious reflections. These data therefore represent a challenging test case for our approach which considers all integrated reflections without outlier rejection.

To address the outliers, we used the cross-validation implemented in Careless to select an appropriate degrees-of-freedom (d.f.)

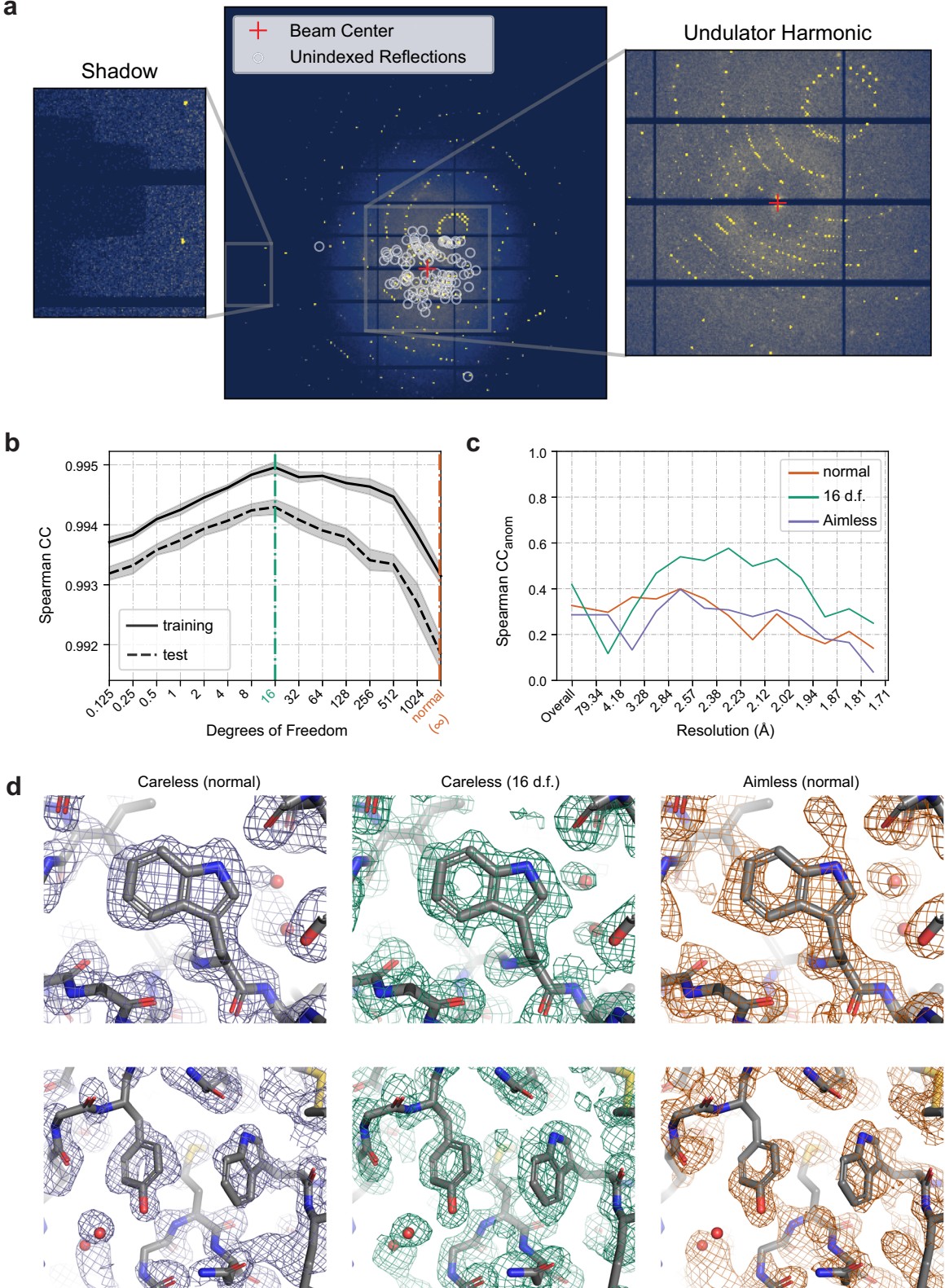

**Fig. 3 | Accurate processing of sulfur SAD data with Careless. a** A sample diffraction pattern from the lysozyme data set indicating strong spots which could not be indexed by DIALS[52]. Insets show sources of outliers in the data (left: a beam stop shadow; right: a secondary diffraction pattern resulting from an undulator harmonic). **b** Ten-fold cross-validation of merging as a function of the likelihood degrees of freedom. Lines: average values; gray bands: bootstrap 95% confidence intervals from 10 repeats with different randomly chosen test reflections. **c** Spearman correlation coefficient between anomalous differences estimated from half-datasets with jointly trained scale function parameters. **d** Density-modified experimental electron density maps produced with PHENIX Autosol[21] using the sulfur substructure from a reference structure (PDBID: 7L84), contoured at 1.0 $\sigma$. Rendered with PyMOL[62].

parameter of a Student's $t$-distributed likelihood function for these data. Titrating the number of degrees of freedom, we found that 16 d.f. resulted in the best Spearman correlation coefficient between observations and model predictions (Fig. 3b). Accordingly, at 16 degrees of freedom, the structure factor amplitudes exhibit comparatively high half-dataset anomalous correlations ($CC_{anom}$Fig. 3c). By the same measure, Careless with this tuned likelihood function recovers significantly more consistent signal at intermediate and high resolution than the conventional merging program, Aimless[20]. A more extensive comparison of Careless' scaling protocols with Aimless (Supplementary Fig. 5) shows that this pattern generally holds, whether quantified by Spearman's anomalous correlation coefficient or weighted or unweighted Pearson correlation coefficients.

SAD phasing provides a further test of the quality of anomalous signal inferred by Careless. We used Autosol[21] to phase our merging results, comparing the Careless output with a Student's $t$-distributed likelihood with ∞ or 16 degrees of freedom to the same data merged by Aimless. In order to ensure a consistent origin, we supplied the sulfur atom substructure from the final refined model (PDBID: 7L84) during phasing. Although we provided the heavy atom substructure in the experiments reported here, we were also able to phase each of these data sets ab initio (Supplementary Table 1). It is evident from the density-modified experimental maps in Fig. 3d that the Careless output with the normally distributed error model (∞ d.f.) is of much lower quality. By contrast, both Careless with 16 d.f. and Aimless produced clearly interpretable experimental maps. We repeated these analyses with data processed entirely in XDS, another conventional program that performs all steps from indexing to merging. As shown in Supplementary Fig. 5, scaling in XDS results in $CC_{anom}$ similar to Aimless. XDS output, followed by ab initio phasing in AutoSol, further results in a similar Figure of Merit and somewhat worse Bayes-CC (the difference is within the estimated error). Consistent with this, the experimental electron density map from XDS (Supplementary Fig. 6a) is of slightly lower visual quality for the scenes illustrated in Fig. 3d.

Anomalous signal in real space, on cysteine and methionine S atoms, provides an additional measure of the accuracy of the estimated structure factor amplitudes. To this end, we performed limited automated refinement in Phenix using a sulfur-omit version of PDB ID 7L84[19] as a starting model, and inferred peak heights from the resulting anomalous omit map. As shown in Supplementary Table 2, Aimless outperforms Careless with 16 d.f. and XDS in this regard, underscoring the subtle differences in the requirements each test imposes on the data.

As we illustrate in the online example "Boosting SAD signal with transfer learning", it is possible to further improve scaling in Careless by using a simple transfer-learning procedure in which the parameters of the scale function are learned by a non-anomalous pre-processing step. With this addition, Careless attains higher average anomalous peak height than Aimless and XDS (Supplementary Table 2), better Spearman $CC_{anom}$ (Supplementary Fig. 5), and equal (by figure of merit) or better (by Bayes-CC and visual appearance) phased map quality (Supplementary Table 1, Supplementary Fig. 6c). Moreover, Careless can post-process already scaled XDS data and improve anomalous peak heights (Supplementary Table 2). Relative performance may, of course, vary from dataset to dataset. In summary, Careless supports the robust recovery of high-quality experimental electron density maps and anomalous signal in the presence of physical artifacts.

### Sensitive detection of time-resolved change from polychromatic diffraction data

Polychromatic (Laue) X-ray diffraction provides an attractive modality for serial and time-resolved X-ray crystallography, as many photons can be delivered in bright femto- or picosecond X-ray pulses[22]. In particular, most reflections in these diffraction snapshots are fully, rather than partially, observed even in still diffraction images. Laue diffraction processing remains, however, a major bottleneck[3] due to its polychromatic nature: The spectrum of a Laue beam is typically peaked with a long tail toward lower energies. This so-called "pink" beam means that reflections recorded at different wavelengths are inherently on different scales. In addition, reflections, which lie on the same central ray in reciprocal space will be superposed on the detector. These "harmonic" reflections need to be deconvolved to be merged[23].

Typical polychromatic data reduction software addresses these issues in a series of steps—it uses the experimental geometry to infer which photon energy contributed most strongly to each reflection observation. It then scales the reflections in a wavelength-dependent manner by inferring a wavelength normalization curve related to the spectrum of the X-ray beam[24]. Finally, it deconvolves the contributions to each harmonic reflection by solving a system of linear equations for each image[23]. The need for these steps made it difficult to scale and merge polychromatic data. Not surprisingly, there are no modern open-source merging packages supporting wavelength normalization and harmonic deconvolution.

By contrast, the forward modelling approach implemented in Careless readily extends to the treatment of Laue diffraction. First, to handle wavelength normalization, providing the wavelength of each reflection estimated from experimental geometry as metadata enables the scale function to account for the nonuniform spectrum of the beam. Harmonic deconvolution requires accounting for the fact that the intensity of a reflection is the sum over contributions from all Miller indices lying on the relevant central ray—in the forward probabilistic model implemented in Careless this is a simple extension of the monochromatic case.

To demonstrate that Careless effectively merges Laue data, we applied it to a time-resolved crystallography data set—20 images from a single crystal of photoactive yellow protein (PYP) in the dark state and 20 images each collected 2 ms after a blue laser pulse. Blue light induces a *trans*- to-*cis* isomerization in the *p*-coumaric acid chromophore in the PYP active site, which can be observed in time-resolved experiments[25] (Fig. 4a). We first integrated the Bragg peaks using the commercial Laue data analysis software, Precognition (Renz Research). Then we merged the resulting intensities using Careless.

Careless produces high-quality structure factor amplitudes for this data set, as judged by half-data set correlation coefficients (Fig. 4c and 4b). We refined a ground-state model against the 'dark' data starting from a reference model (PDBID: 2PHY)[26], yielding excellent $2F_o - F_c$ electron density maps (Fig. 4d). Using the phases from this refined ground-state model, we then constructed unweighted difference maps $|\Delta F_h| \approx (|F_h^{2ms}| - |F_h^{dark}|)$, $\phi_h \approx \phi_{h,calc}^{dark}$. As shown in Fig. 4e, these maps contain peaks around the PYP chromophore. To better visualize these maps, we applied a previously described weighting procedure[27]. The weighted maps (Fig. 4f, 4g) show strong difference density localized to the chromophore, consistent with published models of the dark and excited-state structures[25].

Previous generations of Laue merging software required discarding reflections below a particular $I/\sigma_I$ cutoff during scaling and merging. Otherwise, the resulting structure factor estimates are not accurate enough to be useful in the analysis of time-resolved structural changes. Here, we applied no such cutoff. Likewise, the appearance of interpretable difference electron density in the absence of a weighting scheme (Fig. 4e) is extraordinary. The ability of Careless to identify these difference signals demonstrates an unprecedented degree of accuracy and robustness to outliers. As such, Careless improves on the state of the art for the analysis of Laue experiments.

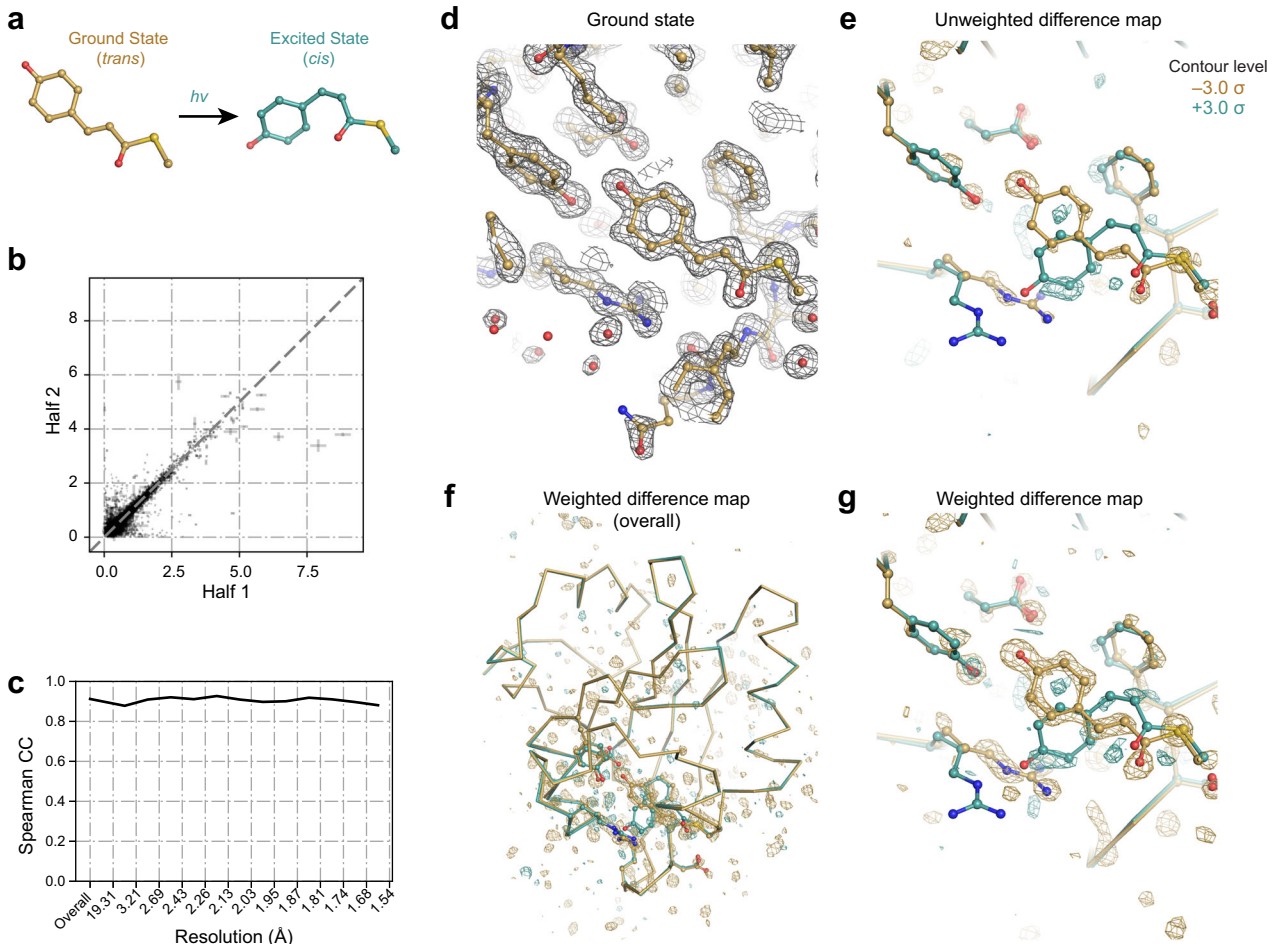

**Fig. 4 | Careless accurately merges time-resolved polychromatic diffraction data. a** When exposed to blue light, the PYP chromophore undergoes *trans*-to-*cis* isomerization. In total, 40 images from a single crystal of PYP were processed: 20 were recorded in the dark state, and 20 were recorded 2 ms after illumination with a blue laser pulse. Each image is the result of 6 accumulated X-ray pulses. The signal to noise and multiplicity of each time point is reported in Supplemental Fig. 4. **b** The data were randomly divided in half by image and merged with scale function parameters learned by merging the full data set. Merging with Careless gave excellent correlation between the structure factor estimates of both halves. Error bars indicate the standard deviation of the structure factor posteriors for the half data sets around the posterior means. **c** Half-dataset correlation coefficients as a function of resolution, including both the dark and 2 ms data. **d** Ground-state $2F_o - F_c$ map created by refining the ground-state model (PDBID: 2PHY) against the dark merging results (countoured at 1.5 $\sigma$). The phases of this refined model were used for the difference maps in (**e-g**). **e** $F^{2ms} - F^{dark}$ time-resolved difference map showing the accumulation of blue positive density around the excited state chromophore (blue model, PDBID: 3UME) and depletion of the ground state (yellow model, PDBID: 2PHY). **f** $F^{2ms} - F^{dark}$ weighted time-resolved difference map showing localization of the difference density to the region surrounding the chromophore. **g** $F^{2ms} - F^{dark}$ weighted time-resolved difference map showing large differences around the chromophore. All difference maps are contoured at ± 3.0$\sigma$.

## Recovering anomalous signal from a serial experiment at an X-ray Free-Electron Laser

X-ray Free-Electron Lasers (XFELs) are revolutionizing the study of light-driven proteins[28–32], enzyme microcrystals amenable to rapid mixing[33–35], and the determination of damage-free structures of difficult-to-crystallize targets[36,37]. Diffraction data from XFEL sources involve two unique challenges that result from the serial approach commonly used to outrun radiation damage[4]. The first challenge of serial crystallography is that each image originates from a different crystal with a different scattering mass, which diffracts one intense X-ray pulse before structural damage occurs. A completely global scaling model is therefore not appropriate. To overcome this limitation, we exploited the modular design of Careless to incorporate local parameters into the scale function. Specifically, we appended layers with per-image kernel and bias parameters to the global scale function (Fig. 5a). Effectively, these additional layers allow the model to learn a separate scale function for each image. To address the risk of overfitting posed by the additional parameters, we determined the optimal number of image layers by crossvalidation (see Supplementary Fig. 4

for determination of optimal the number of image-specific layers). A second challenge of serial crystallography is that all images are stills— there is neither time to rotate the crystal during exposure, nor the spectral bandwidth to observe the entirety of each reciprocal lattice point (Fig. 5b). For a still image, the maximal intensity for a given reflection is observed on the detector when the so-called Ewald sphere intersects the reflection centroid. The Ewald offset (EO) measures the degree to which a particular reflection observation deviates from its maximal diffraction condition (the length of the orange arrow in Fig. 5b) and can be estimated from the experimental geometry for each observation. To account for partiality, we hence provided the EO estimates as a metadata.

To test if our model could leverage per-image scale parameters and EO estimates, we applied Careless to unmerged intensities from an XFEL serial crystallography experiment (CXIDB, entry 81). In this experiment[38], a slurry of thermolysin microcrystals was delivered to the XFEL beam by a liquid jet. The data contain significant anomalous signal from the zinc and calcium ions in the structure. We limited analysis to a single run

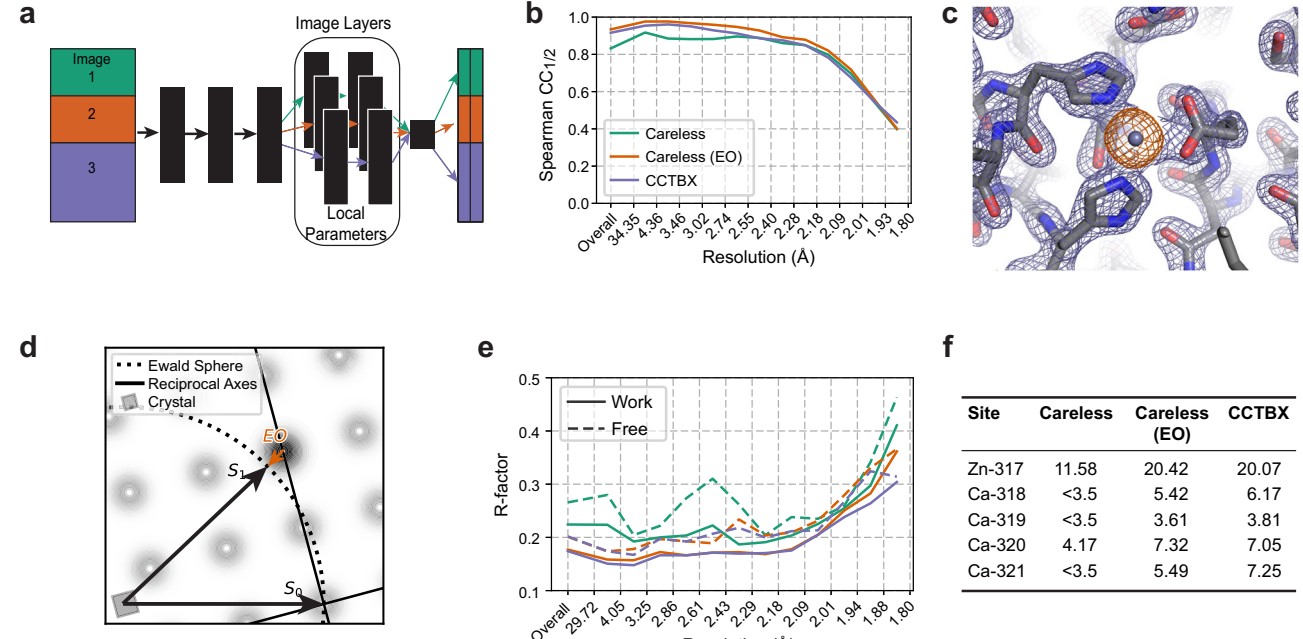

**Fig. 5 | Careless can leverage geometric metadata to improve XFEL data processing. a** By adding image-specific layers to the neural network, Careless can scale diffraction data from serial XFEL experiments. **b** The Ewald offset (EO), shown in red, can optionally be included during processing. $S_0$ and $S_1$ represent the directions of the unscattered and diffracted X-rays, respectively. **c** Half-dataset correlation coefficients by resolution bin for data processed in Careless, with and without Ewald offsets, and CCTBX. **d** Refinement R-factors from phenix.refine using Careless, with and without Ewald offsets, and CCTBX. **e** Careless $2F_o - F_c$ electron density map, from inclusion of EO, contoured at 2.0 $\sigma$ (purple mesh) overlayed with thermolysin anomalous omit map contoured at 5.0 $\sigma$ (orange mesh). **f** Peak heights of anomalous scatterers in an anomalous omit map, in $\sigma$ units, for Careless output with and without Ewald offsets and conventional data processing with CCTBX.

containing 3,160 images. We processed the integrated intensities in Careless using ten global layers and two image layers, both with and without the inclusion of the Ewald offset metadata to evaluate whether its inclusion improves the analysis.

As shown in Fig. 5, Careless successfully processes the serial XFEL data. In particular, use of the EO metadatum yielded markedly superior results as judged by the half-dataset correlation coefficient (Fig. 5c) as well as the refinement residuals (Fig. 5d). To verify that including the metadata also improved the information content of the output, we constructed anomalous omit maps after phasing the merged structure factor amplitudes by isomorphous replacement with PDBID 2TLI[39]. Specifically, we omitted the anomalously scattering ions in the reference structure during refinement against the Careless output. The anomalous difference peak heights at the former location of each of the anomalous scatterers, tabulated in Fig. 5f, confirm that the inclusion of Ewald offsets not only improved the accuracy of structure factor amplitudes estimates but also yielded anomalous differences on par with XFEL-specific analysis methods.

## Discussion

### Statistical modeling can account for diverse physical effects
We have shown that Careless successfully scales and merges X-ray diffraction data without the need for explicit physical models of X-ray scattering—comparing favorably to established algorithms tailored to specific crystallography experiments. This begs the question: What sorts of physical effects can the Careless scale function account for using a general-purpose neural network and Wilson's prior distributions on structure factor amplitudes[15]? Though our scale function operates in a high-dimensional space, making it difficult to interrogate directly, we have found that the impact of excluding specific reflection metadata can be used to assess the physical effects that are implicitly accounted for by Careless.

For example, the resolution of each observed reflection is essential for proper data reduction in Careless. This suggests that, among other corrections, the model learns an isotropic scale factor akin to the per-image temperature factors included in most scaling packages[6]. We have also found that in some cases the merging model benefits from the inclusion of the observed Miller indices during scaling, as was the case with the lysozyme and PYP data presented here. This indicates that for some data sets Careless learns an effective anisotropic scaling model. Likewise, the inclusion of the detector positions of the reflections seems to improve merging performance. Since no prior polarization correction has been applied to the PYP and thermolysin intensities used here, this suggests that source polarization is implicitly corrected by the model. Specific crystallographic experiments can also benefit from the inclusion of domain-specific metadata. The success of merging Laue data implies that Careless can learn a function of the spectrum of the X-ray source (Fig. 4). Finally, the XFEL example shows the model is competent to infer partialities in still images (Fig. 5). In principle, the inclusion of local parameters in the form of image layers allows the model to make geometric corrections similar to the partiality models implemented in other packages[40–42]. The major difference is that our work does not require an explicit model of the line shape of Bragg peaks nor of the resolution-dependence of the peak size (mosaicity).

### Robust statistics instead of outlier rejection
Occasionally, observed reflections have spurious measured intensities. These outliers can arise from various physical effects during a diffraction experiment, such as ice or salt crystals, detector readout noise, or absorption and scattering by surrounding material[6]. In conventional data reduction, outlier observations are frequently detected and filtered during data processing to improve the estimates of merged intensities and statistics[6,13]. This step is necessary because the inverse-variance weighting scheme[43] is otherwise easily skewed by spurious observations.

Instead of outlier rejection, Careless employs robust statistical estimators. The processing of the native SAD data from lysozyme

(Fig. 3) show its benefit. These data have significant outliers at low and high resolution (Fig. 3a), and the anomalous content is notably improved through the use of a robust error model (Fig. 3c). Importantly, doing so also improves experimental electron density maps compared to the corresponding normally distributed error models (Fig. 3d). The influence of outlier observations can be tuned using the degrees of freedom of the Student's $t$-distribution during data processing. This approach to handling outlier observations highlights the flexibility of the Careless model and reduces the need for data filtering steps.

### Sensitive detection of small structural signals in change-of-state experiments

Our processing of the time-resolved PYP photoisomerization Laue dataset demonstrates that Careless can accurately recover the signal from small structural changes (Fig. 4). The use of Careless in this context involved fitting a common scaling model for both the dark and 2ms datasets, while inferring two separate sets of structure factor amplitudes, one for each state. The quality of the difference maps generated from this processing (Fig. 5e-g) suggests that the common scaling model employed by Careless is effective for analyzing these change-of-state experiments. Importantly, the common scaling model may improve difference maps by ensuring that inferred structure factor amplitudes are on the same scale, producing a balanced difference map with both positive and negative features.

These features suggest that Careless will have strong applications in time-resolved experiments and related change-of-state crystallography experiments. Furthermore, although Careless currently only provides Wilson's distributions as a prior over structure factor amplitudes, it is possible to imagine using stronger priors to further constrain the inference problem. Such priors are an active area of research, and further improve the sensitivity to small differences between conditions in time-resolved data sets (see the online example "Using a bivariate prior to exploit correlations between Friedel mates" for a prototype implementation).

### Supporting next-generation diffraction experiments

In its current implementation, Careless requires that the full data set reside in memory on a single compute node or accelerator card (each presented examples can run on a consumer-grade NVIDIA 3000 series GPU in under an hour). This is a significant limitation for free-electron laser applications. The next generation of X-ray free-electron lasers will provide data acquisition rates of $10^3 - 10^6$ diffraction images per second[44] leading to very large data sets with images numbering in the millions or more. In this setting, it would be advantageous to have an online merging algorithm, which did not require access to the entire data set at each training step. Currently, the reliance on local parameters to handle serial crystallography data precludes this. However, we are exploring strategies to replace these local parameters with a global function. This will enable Careless to be implemented with stochastic training[45], in which gradient descent is conducted on a subset of the data at each iteration. This training paradigm allows variational inference to be used for data sets too large to fit in memory. With stochastic training, variational inference will be an excellent candidate for merging large XFEL data sets on-the-fly during data acquisition.

In summary, we have described a general, extensible framework for the inference of structure factor amplitudes from integrated X-ray diffraction intensities. We find the approach to be accurate, and applicable to a wide range of X-ray diffraction modalities. Careless is modular and open-source. We encourage users to report their experiences with downstream software and to contribute extensions through github.com/rs-station/careless. Careless provides a foundation for the ongoing development and systematic application of

advanced probabilistic models to the analysis of ever more powerful diffraction experiments.

## Methods

### Merging X-ray data by variational inference

Observed reflection intensities can be thought of as the product of diffraction in an ideal experiment and a local scale, which describes the systematic error in each reflection observation. This implies a graphical model (Supplementary Fig. 1), relating the observed intensities, $I_{h,i}$ to two sets of latent variables, the reflection scales, $\Sigma_{h,i}$, and the structure factor amplitudes, $F_h$. The corresponding joint distribution factorizes as

$$p(I,F,\Sigma) = p(I|F,\Sigma)p(F)p(\Sigma). \tag{1}$$

In this setting, it is most desirable to estimate the posterior,

$$p(F,\Sigma|I) = \frac{p(I,F,\Sigma)}{p(I)} \tag{2}$$

$$= \frac{p(I|F,\Sigma)p(F)p(\Sigma)}{p(I)}. \tag{3}$$

The exact posterior is generally intractable for such problems. So, we posit an approximate posterior $q$ taken from a parametric family of distributions. This is the so-called variational distribution or surrogate posterior. We then use optimization to learn parameters of $q$ such that it approximates the desired posterior. One way to accomplish this is to minimize the Kullback-Leibler divergence between $q$ and the posterior.

$$D_{KL}[q \parallel p(F,\Sigma|I)] = \mathbb{E}_q\big[\log q - \log p(F,\Sigma|I)\big] \tag{4}$$

$$= \mathbb{E}_q\big[\log q - \log p(I|F,\Sigma) - \log p(F) - \log p(\Sigma) + \log p(I)\big] \tag{5}$$

Note that the expectation, $\mathbb{E}_q\big[\log p(I)\big]$ does not depend on the parameters of $q$. It is therefore a constant. Disregarding this constant term,

$$D_{KL}[q \parallel p(F,\Sigma|I)] \propto \mathbb{E}_q\big[\log q - \log p(I|F,\Sigma) - \log p(F) - \log(\Sigma)\big], \tag{6}$$

and negating,

$$\text{ELBO}(q) = \mathbb{E}_q\big[-\log q + \log p(I|F,\Sigma) + \log p(F) + \log(\Sigma)\big], \tag{7}$$

leads to the optimization objective of variational inference, which is called the Evidence Lower BOund (ELBO)[17,18]. Maximizing this quantity with respect to the variational distribution, $q$, recovers an approximation to the posterior distribution. After re-arranging the terms,

$$\text{ELBO}(q) = \mathbb{E}_q\big[\log p(I|F,\Sigma) - \log q + \log p(F) + \log p(\Sigma)\big] \tag{8}$$

$$= \mathbb{E}_q\big[\log p(I|F,\Sigma)\big] - \mathbb{E}_q\big[\log q - \log p(F) - \log p(\Sigma)\big] \tag{9}$$

$$= \mathbb{E}_q\big[\log p(I|F,\Sigma)\big] - D_{KL}(q \parallel p(F)p(\Sigma)), \tag{10}$$

it is clear that the ELBO can be thought of as the sum of expected log-likelihood of the data and the negative Kullback-Leibler divergence between the surrogate posterior and the the prior. The expected log-likelihood term encourages the model to faithfully represent the data. The KL divergence term acts as a penalty, which discourages the surrogate posterior from wandering too far from the prior distribution. From the frequentist perspective, this is similar to a regularized maximum-likelihood estimator. This general form of the ELBO applies

equally to any parameterization of the graphical model in Supplementary Fig. 1.

The parameterization used in this work slightly simplifies Equation (10). The graphical model in Supplementary Fig. 1, implies that the prior distribution $p(F, \Sigma)$ factorizes $p(F)p(\Sigma)$. Therefore, it is convenient to assume the surrogate posterior,

$$q(F,\Sigma) = q_F(F)q_\Sigma(\Sigma), \tag{11}$$

consists of statistically independent distributions for $F$ and $\Sigma$. This is a modeling choice, and it need not be the case in other variational merging models with this graph. Factorizing $q$ leads to an ELBO,

$$\text{ELBO}(q) = \mathbb{E}_q\left[\log p(I|F,\Sigma)\right] - D_{KL}\left(q_F \parallel p(F)\right) - D_{KL}\left(q_\Sigma \parallel p(\Sigma)\right), \tag{12}$$

with separate Kullback-Leibler divergences for $F$ and $\Sigma$. We can now begin to consider priors for each of these surrogate posteriors.

## An uninformative prior on scales

It is difficult to reason about the appropriate prior distribution for scales, $p(\Sigma)$. In all likelihood, this prior depends intimately on the details of the experiment. It will vary by sample and apparatus. In this work we choose an uninformative prior, $\Sigma \sim q(\Sigma)$. Thereby, the second divergence term in the ELBO becomes zero, and whatever parameters define $q_\Sigma$ are simply allowed to optimize as dictated by the likelihood term. The objective used in this work is therefore

$$\text{ELBO}(q) = \mathbb{E}_q\left[\log p(I|F,\Sigma)\right] - D_{KL}\left(q_F \parallel p(F)\right). \tag{13}$$

## Posterior structure factors

In this work, the surrogate posteriors of structure factors are independently parameterized by truncated normal distributions.

$$q_{F_h}(F) = \text{TruncatedNormal}\left(F|\mu_{F_h}, \sigma_{F_h}\right) \tag{14}$$

with location and scale parameters $\mu_{F_h}$ and $\sigma_{F_h}$ and support

$$F \in \begin{cases} [0,\infty) & \text{centric} \\ (0,\infty) & \text{acentric}. \end{cases} \tag{15}$$

Both the location and scale parameters are constrained to be positive. This constraint is implemented with the softplus function, $\text{softplus}(x) = \log(\exp(x)+1)$ in Careless version 0.2.0 and with the exp in subsequent versions including 0.2.3 used for the additional lysozyme analyses in the supplemental information.

## Posterior scales

As noted in section 'An Uninformative Prior on Scales', we choose not to impose a structured prior on reflection scales. Rather, we assert that the scale of a reflection should be computable from the geometric metadata recorded about each reflection during integration. Therefore, our model infers a function, which ingests metadata and outputs scale factors (Fig. 2a).

By default, we parameterize this function as a deep neural network with parameters $\theta$. In particular, we use a multilayer perceptron with leaky rectified linear units (ReLU) as the activation. The parameters, correspond to the kernels and biases of each layer. The kernels are initialized to the identity matrix and biases to zeros. The number of hidden units in each layer takes the dimensionality of the metadata by default, but this is user-configurable. We set the default depth of the neural network to twenty layers which we find offers a reasonable balance of performance and stability. The model has a final, linear layer with 2 units. The output of the last layer is interpreted as the mean and standard deviation of the scale distribution with the standard

deviation constrained to be positive in the same manner as the structure factor posteriors' parameters .

$$\left\{\mu_{\Sigma_{h,i}}, \sigma_{\Sigma_{h,i}}\right\} = f_\theta(\mathbf{M_{h,i}}) \tag{16}$$

$$q_{\Sigma_{h,i}} = \text{Normal}\left(\mu_{\Sigma_{h,i}}, \sigma_{\Sigma_{h,i}}\right) \tag{17}$$

The correct scale function is the one which allows the model to recapitulate the data while letting the structure factors follow the desired prior distribution. Provided rich enough metadata about each reflection observation, variational inference will recover such a function.

One must use caution when selecting metadata. If information about the reflection intensities is provided to the scale function, the scale function may bypass the structure factors to directly minimize the expected log-likelihood. This leads to poor structure factor estimates. We recommend against including data such as the reflection uncertainties in the metadata, as they are strongly correlated with the intensities.

## Wilson's Priors

Wilson's priors,

$$p(F_h) = \text{Wilson}(F_h) = \begin{cases} \text{Halfnormal}(F_h|\epsilon_h) = \sqrt{\frac{2}{\pi\epsilon_h}}\exp\left(-\frac{F_h^2}{2\epsilon_h}\right) & h \text{ is centric} \\ \text{Rayleigh}(F_h|\epsilon_h) = \frac{2}{\epsilon}F_h\exp\left(-\frac{F_h^2}{\epsilon_h}\right) & h \text{ is acentric}, \end{cases} \tag{18}$$

express the expected distributions of structure factors under the assumption that atoms are uniformly distributed within the unit cell[15]. The probability distribution over the structure factor amplitude $F_h$ with Miller index $h$ is expressed in terms of the multiplicity of the reflection $\epsilon_h$. The multiplicity, a feature of the crystal's space group, is a constant which can be determined for each Miller index. It corresponds to the contribution to the relative intensity of each reflection solely due to crystal symmetry. The Wilson prior has separate parameterizations for centric and acentric reflections. This form of Wilson's priors differs from the one employed in the French Wilson algorithm[14] in that it is independent of the scale, $\Sigma$. Because of this choice, the scale function can be inferred in parallel with the structure factor amplitudes. However, it implies that the structure factors output by Careless are on the same scale across resolution bins. This is an important consideration for downstream processing. Careless output may, for some applications, need to be rescaled to meet the expectations of crystallographic data analysis packages. An input flag is available to apply a global Wilson B-factor.

## Likelihood functions

The first term in the ELBO is an expected log-likelihood. In this work, we present two parameterizations of this term: a normal distribution

$$p(I_{h,i}|F_h,\Sigma_{h,i}) = \text{Normal}(I_{h,i}|F_h^2\Sigma_{h,i}, \sigma_{I_{h,i}}), \tag{19}$$

which is suitable for data with few outliers, as well as a robust t-distribution

$$p(I_{h,i}|F_h,\Sigma_{h,i}) = \text{StudentT}(I_{h,i}|\nu, F_h^2\Sigma_{h,i}, \sigma_{I_{h,i}}), \tag{20}$$

which adds an additional hyperparameter. The degrees of freedom, $\nu$, titrates the robustness of the model toward outliers. In the limit $\nu \to \infty$, the t-distribution approaches a normal distribution.

## Model training

In this work, we use the reparameterization trick to estimate gradients of the ELBO with respect to the parameters of the variational distributions, $q$. First applied in the Variational Autoencoder[46],

reparameterization is a common tool to estimate gradients of probabilistic programs. In our implementation, the ELBO is approximated by random samples from the surrogate distributions,

$$\text{ELBO}\left(\mu_{q_F}, \sigma_{q_F}, \theta\right) = \mathbb{E}_q \left[\log p\left(I|F^2\Sigma\right)\right] - D_{KL}(q_F \parallel p(F)) \quad (21)$$

$$\approx \sum_{s=1}^{S} \left[ \sum_i \sum_h \log p(I_{h,i}|F_{h,s}^2 \Sigma_{h,i,s}, \sigma_{I_{h,i}}) - \sum_h \left(\log q_{F_h}(F_{h,s}) - \log p(F_{h,s})\right) \right] \quad (22)$$

where $F_{h,s}$ and $\Sigma_{h,i,s}$ denote reparameterized samples from the surrogate posteriors, and the number of Monte Carlo samples, $S$ is a hyperparameter. By default a single sample is used ($S=1$). For training, we use the Adam optimizer[47] with hyperparameters $\alpha = 0.001$, $\beta_1 = 0.9$, and $\beta_2 = 0.99$.

## Cross-validation

Careless provides two modes of cross-validation. In the first paradigm, the model is first trained on the full data set yielding structure factor estimates and neural network weights. Next, the data are partitioned randomly into halves by image. Using the neural network weights learned from the full data set, each half is merged separately by optimizing the structure factors. During this process the neural network weights remain fixed. The resulting pair of structure factor estimates may be correlated to produce a measure similar to the canonical $CC_{1/2}$ widely used in crystallography. This mode of cross-validation does not necessarily inform the user about the degree of overfitting. Rather, the $CC_{1/2}$ value is more indicative of the data consistency.

The second type of cross-validation supported by Careless is intended to explicitly test for overfitting in the scale function. In this mode, a fraction of the data is held out during training time. After training, the model is applied to these data in order to predict intensities for the held-out fraction. The correlation between the observed intensities and the predictions provides an estimate for how well the model generalizes. The choice of summary statistic is up to the user. However, we recommend Spearman's rank correlation coefficient as a robust alternative to Pearson's. In the following section, we address the issue of how to recover intensity predictions and moments from our model.

## Predictions

Model predictions are essential to quantify model overfitting by cross-validation. The predicted reflection intensities implied by our model are the product of random variables,

$$\hat{I}_{h,i} = \Sigma_{h,i} F_h^2, \quad (23)$$

The variational distributions inferred by Careless imply a probability distribution for the intensity of each reflection. We do not have an analytical expression for this distribution but can compute the expected value,

$$\langle \hat{I}_{h,i}\rangle = \langle\Sigma_{h,i}\rangle\langle F_h^2\rangle \quad (24)$$

$$= \langle\Sigma_{h,i}\rangle\left(\langle F_h\rangle^2 + \sigma_{F_h}^2\right), \quad (25)$$

taking advantage of the fact that in the Careless formalism $\Sigma_{h,i}$ and $F_h$ are assumed to be statistically independent in the posterior distribution. The first term in the product,

$$\langle\Sigma_{h,i}\rangle = \mu_{\Sigma_{h,i}} \quad (26)$$

is computed by the scale function, $f_\theta$, from the metadata vector, $\mathbf{M_{h,i}}$,

$$\left\{\mu_{\Sigma_{h,i}}, \sigma_{\Sigma_{h,i}}\right\} = f_\theta(\mathbf{M_{h,i}}). \quad (27)$$

Note that $f_\theta$ returns a two-element vector, the first of which is the expected value of $\Sigma_{h,i}$, and the second is the standard deviation. The second term, is calculated from the moments of the truncated normal surrogate posterior, $q_{F_h}$. These moments have analytical expressions which are implemented in many statistical libraries, including TensorFlow Probability[48] which we use in this work.

It is also possible to compute the second moment of the predictions,

$$\sigma_{I_{h,i}}^2 = \langle\hat{I}_{h,i}^2\rangle - \langle\hat{I}_{h,i}\rangle^2 \quad (28)$$

$$= \langle F_h^4\rangle\langle\Sigma_{h,i}^2\rangle - \langle\hat{I}_{h,i}\rangle^2 \quad (29)$$

$$= \langle F_h^4\rangle\left(\sigma_{\Sigma_{h,i}}^2 + \mu_{\Sigma_{h,i}}^2\right) - \langle\hat{I}_{h,i}\rangle^2, \quad (30)$$

where the fourth non-central moment, $\langle F_h^4\rangle$ of $q_{F_h}$ has an analytical expression which is implemented in SciPy[49].

## Harmonic deconvolution for Laue diffraction

To implement harmonic deconvolution, the Careless ELBO approximator needs to be modified to update the center of the likelihood distribution. By summing over each contributor on the central ray, the new ELBO approximation becomes

$$\text{ELBO}\left(\mu_{q_F}, \sigma_{q_F}, \theta\right) \approx$$
$$\sum_{s=1}^{S} \left[ \sum_i \sum_{ray} \log p\left(I_{ray,i}| \sum_{h \in ray}\left(F_{h,s}^2 \Sigma_{h,i,s}\right), \sigma_{I_{ray,i}}\right) - \sum_h \left(\log q_{F_h}(F_{h,s}) - \log p(F_{h,s})\right) \right] \quad (31)$$

which is readily optimized by same protocol demonstrated in Supplementary Fig. 3. In the code base, harmonics are handled by having a separate class of likelihood objects for Laue experiments. In practice, one could use the polychromatic likelihood to merge monochromatic data with no ill effect on the quality of the results. In that sense, this is the more general version of the ELBO for diffraction data. However, doing so would incur a performance cost given the underlying implementation, which is why we maintain separate likelihoods for mono and polychromatic experiments. Regardless, the core merging class inside Careless is competent to fit both sorts of data.

## Data collection and analysis

Data collection for hen egg white lysozyme was described in reference[19]. Data for photoactive yellow protein were collected as described in reference[50]. Collection of thermolysin data was described in[38]. Scaling and merging was performed using Careless version 0.2.0 or version 0.2.3 in the case of the "Image Layer" and "Transfer" protocols presented in Supplementary Tables 1 and 2 and Supplementary Figs. 5 and 6. DIALS version 3.1.4 was used to index and integrate observed reflections for hen egg white lysozyme. Aimless version 0.7.4 was used to merge the integrated intensities for hen egg white lysozyme data. For the XDS analysis, the Jan 10, 2022 version was used along with the 2021/1 version of SHELX. Precognition version 5.2.2 was used to index and integrate the polychromatic PYP data. cctbx.xfel version 2021.11.dev3+4.g05389c3054 was used to scale and merge thermolysin XFEL data; the merging parameters are available in the Zenodo deposition. All model refinement and phasing was performed in PHENIX version 1.18.2. The refinement outputs and log files, including parameter settings are deposited in Zenodo. The

anomalous peak heights presented in Fig. 5 and Supplementary Table 2 report anomalous signal based on anomalous difference maps obtained after refinement of an S-omit (Supplementary Table 2) or Zn, Ca-omit refined model of each protein, and were quantified using the "Difference map peaks…" function Coot version 0.9.6. For Fig. 3, we used five cycles of isotropic atomic temperature factors and rigid body refinement. For thermolysin, Fig. 5, atomic coordinates were also refined as they improved refinement residuals. One could, instead, work with initial omit map peak heights reported during de novo phasing. In our experience, these anomalous peak heights depend strongly on substructure search parameters, and are therefore less suitable for comparison of scaling methods.

### Reporting summary

Further information on research design is available in the Nature Portfolio Reporting Summary linked to this article.

## Data availability

The refined lysozyme structure[19] is deposited in the Protein Data Bank under PDBID 7L84 [https://www.rcsb.org/structure/7L84]. The raw lysozyme diffraction images are available through the SBGrid Data Bank under accession code 816 [https://data.sbgrid.org/dataset/816/]. The ground state photoactive yellow protein structure[26] is deposited in the Protein Data Bank under 2PHY [https://www.rcsb.org/structure/2PHY]. The excited state photoactive yellow protein structure[60] is deposited in the Protein Data Bank under 3UME [https://www.rcsb.org/structure/3UME]. The thermolysin structure[39] is deposited in the Protein Data Bank under 2TLI [https://www.rcsb.org/structure/2TLI]. The integrated thermolysin diffraction data[61] are available from the Coherent X-ray Imaging Data Bank under accession code 81 [http://www.cxidb.org/id-81.html]. Some data processing statistics are provided in Supplementary Tables 3–5 as well. The three data sets discussed in this manuscript have been adapted into examples available through the `careless-examples` GitHub page [https://github.com/rs-station/careless-examples] including the unmerged diffraction data and relevant merging scripts. The results presented here including all merged structure factors and crossvalidation data along with the intermediate analysis used to generate all figures and tables had been deposited in Zenodo under accession code 10.5281/zenodo.6408749 [https://doi.org/10.5281/zenodo.6408749].

## Code availability

The source code used to generate all figures and tables is freely available from Zenodo [https://doi.org/10.5281/zenodo.6408749]. The algorithm described, Careless, is implemented in a python package which is available from our GitHub page (https://github.com/rs-station/careless). It can be installed on Mac OS or Linux with the popular python package manager, `pip`.

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

## Acknowledgements
We acknowledge the open source software projects used in this work including Daft[51], DIALS[52], GEMMI[53], Matplotlib[54], Numpy[55], Pandas[56], reciprocalspaceship[57], SciPy[49], Seaborn[58], TensorFlow[59], TensorFlow Probability[48]. We thank Vukica Šrajer and Marius Schmidt for providing Laue time-resolved data, Nick Sauter and Aaron Brewster for helpful discussions, and T.J. Lane and Takanori Nakane for comments on the manuscript. We thank Derek Mendez for advice on CCTBX. We thank the staff at the Northeastern Collaborative Access Team (NE-CAT), beamline 24-ID-C of the Advanced Photon Source for assistance with room-temperature crystallography, in particular Igor Kourinov. NE-CAT beamlines are supported by the National Institute of General Medical Sciences, NIH (P30 GM124165), using resources of the Advanced Photon Source, a U.S. Department of Energy (DOE) Office of Science User Facility operated for the DOE Office of Science by Argonne National Laboratory under Contract No. DE-AC02-06CH11357. This work was supported by the Searle Scholarship Program (SSP-2018-3240), a Merck Fellowship (338034) from the George W. Merck Fund of the New York Community Trust, and the NIH457 Director's New Innovator Award (DP2-GM141000) to D.R.H. J.B.G. was supported by the National Science Foundation Graduate Research Fellowship under Grant No. DGE1745303. K.M.D. holds a Career Award at the Scientific Interface from the Burroughs Wellcome Fund.

## Author contributions
K.M.D. and D.R.H. conceived the project. K.M.D. conceived and implemented the inference algorithm. J.B.G. contributed to software development. K.M.D. drafted the manuscript. All of the authors contributed to analysis and edited the paper.

## Competing interests
The authors declare no competing interests.
