## [Peer Review File · Nature Communications]

Reviewers' comments:

Reviewer #1 (Remarks to the Author):

This paper describes a procedure for merging X-ray diffraction measurements and for correcting them for systematic errors using a machine-learning approach. The basis for scaling is the same as is used in many merging procedures: measurements are affected by factors that vary slowly in reciprocal space or with other geometrical or time-based parameters, and equivalent measurements are expected to have the same values. The main difference is the use of a machine-learning approach to identify and correct for these factors. A second important difference is that this procedure estimates a Bayesian posterior value that includes prior distribution information, similar to incorporating a French and Wilson calculation in the procedure.

1. The general idea for this procedure seems fine. Machine-learning should be a fine approach for discovering the appropriate scale factors for correction of X-ray diffraction measurements.
2. The procedure has the advantage that it is relatively general and does not require specification of the form of scale factors.

This advantage is somewhat tempered by the fact that existing methods also do not normally require such specification...rather the assumption is often simply that scale factors vary slowly with position in reciprocal space and with time.

3. One concern with the Careless procedure is that the Bayesian posterior estimates that it produces are not well-suited for many of the downstream applications of the X-ray data.

Bayesian methods may give biased results: for example the French and Wilson calculation systematically overestimates values of weak amplitudes for example. In downstream calculations, use of French and Wilson-corrected data can potentially affect estimates of parameters such as the Wilson ratio in twinning.

This effect is already known in molecular replacement and refinement where it is already recommended that intensities be supplied (before a French and Wilson calculation) and not Bayesian estimates of amplitudes. The idea of the Careless procedure is still fine, but introducing it as is and without making it clear what it is suitable for could actually make some aspects of data analysis worse.

4. A second concern is that although the procedure seems fine in principle, it is not clear that it actually works any better than existing methods.

Fig. 3d does indicate that the density-modified map obtained with phenix autosol using data produced by Careless with 16 degrees of freedom is better than the map produced with data

produced by Aimless. However the anomalous peak heights for the same data shown in Supp. Fig. 2 indicate in contrast that the data from Aimless are more accurate. Additionally, the Careless analysis required optimization of a parameter, the number of degrees of freedom to use.

Similarly, Table 5f indicates that the Careless procedure is not better than the CCTBX procedure.

Reviewer #2 (Remarks to the Author):

The paper reports the structure of a membrane protein complex (SR11:HtrII) obtained by serial X-ray crystallography at the SLS synchrotron. Data scaling and merging used a program called Careless which implements a novel algorithm based on a Bayesian framework and Machine Learning, as an alternative to existing programs like e.g. Partialator from the CrystFEL package. The authors claim improvement of data and electron density maps, and build 5 additional residues near the N-terminus of HtrII in a region that appears disordered in other structures of the same complex.

In the Abstract, “more ordered” should be “more ordered compared with structures obtained at cryo temperatures”.

In the Introduction, page 4 top: the sentence “Whereas the transducer protein was truncated to enable crystallization with HtrII residues 1 to 114 being expressed, ...” is difficult to understand. Which residues were truncated? Truncation does not appear to affect the given numbering of 1-114. Or has some renumbering of residues been performed? Is truncation at all relevant for this sentence?

page 4 middle: typo “XELs”

For the sentence “Novel data-processing approaches were also developed to merge very large data-sets that often consisted of many thousands of diffraction images (22).” a single reference is not enough. Other approaches should be cited as well. Similarly, the next sentences make rather general statements but exclusively cite specific applications and papers whose author lists overlap with the current authors. Self citations should at least be balanced against the work of others.

Page 5: “produce significant volumes” should be “produce sufficient number of” or similar.

In Results (Table 1): The overall values of $CC_{1/2}$ and I/σ given here are low and very low, respectively, and the overall value of $CC_{1/2}$ has anyway little significance because it is biased by the falloff of intensity values with increasing resolution. With such atypically poor data (values that are more typical for such experiments, indicate useful data, and convey detail of the high resolution shell, can e.g. be found in Mehrabi et al. (2021)

<https://doi.org/10.1126/sciadv.abf1380> .), structure analysis is not meaningful beyond possibly the level of analysis of domain motions. For the other two entries of the table, the following holds: CC^* is calculated from $CC_{1/2}$ and thus carries no new information, and R_{split} is not given.

Such a table should also detail high-resolution values for R_{split} , I/σ , $CC1/2$ (or CC^*) but these are not shown. Therefore the quality of the data is not only poor, but also remains poorly specified which is unexpected for a paper that claims superior data.

The narrative on page 6 does not give details that go beyond Table 1, and appears to say that a low value of I/σ (0.94) is good! The R_{work} and R_{free} values given here (29 and 34%, respectively, whereas Table 1 has 29.8 and 35.0% !) are high which confirms a situation of poor data in combination with a molecular replacement solution that is based on a generally correct high-resolution model (1H2S).

The following paragraphs focus on improved appearance and interpretation of electron density due to the use of Careless. However, the results are not conclusive. For example, if the backbone density of the traditionally-scaled lower-resolution map is more continuous than the Careless-map (page 7, top), then this reveals the opposite of an improvement due to Careless use. Continuous density at 0.8 sigma for residues 17-21 (why do the authors write "21 to 17"?) in the composite omit map does not prove much, because 0.8 sigma is below the overall level of variation of the map. The electron density of these residues is described as "well defined" (page 8, top) but, as Fig 4d shows, is not continuous in the backbone and shows no sidechains. This density could e.g. result from Fourier ripples at the end of the helix, in combination with phase errors.

Figure 3 shows the $2F_{obs}-F_{calc}$ maps from traditionally-scaled and Careless-scaled data in panels a/b and c/d. Since the maps from Careless-scaled data look better, the authors claim improvement due to the Bayesian treatment. However, the comparisons are not fair: a) the maps are at different resolutions (3.4 vs 2.85Å), and b) the maps suffer from model bias which can be expected to be strong given the high R_{work}/R_{free} values.

I therefore do not agree with the authors' suggestion that "Bayesian statistics deep-learning algorithms (35) provide a powerful tool for improving the quality of serial crystallography data at intermediate resolution."

The sentence "An important advantage to emerge from serial crystallography data collection is the very high multiplicity which results as thousands of partial diffraction patterns are merged into a complete data- set." reflects a misunderstanding: the possibility of merging of data sets requires low systematic differences between them; multiplicity alone is of no use. It is difficult to prove in serial crystallography that the observed differences are actually low; one possibility is refinement of a model to R-values that are similarly low as those obtained in single-crystal work. If this is not proven, merging of (no matter how multiple) diffraction patterns is a recipe for obtaining a set of amplitudes that don't correspond to a single model. What kinds of electron density artifacts this produces depends on the specific case, but it is possible to obtain weak additional density that is interpreted as protein structure, or as in this case, a few additional residues.

Fig. 5 should give references to the structures in the two tables.

Language remark: the authors should re-think their use of the word "recover" (6 times) which I don't find very fitting (see also the online Merriam-Webster Dictionary). E.g. Fig. 2d legend: "recovered" should be "obtained", and Page 5: "were recovered within a lipidic cubic phase" should be "were grown and measured within a lipidic cubic phase", or similar.

Assessment

=====

The conclusions of the paper concerning the “Bayesian statistics founded deep learning algorithm Careless” should be treated with extreme care, as they are based on doubtful interpretations (“improvements”) of properties of the electron density map. In my eyes, the evidence provided does not support the claims, and the paper is therefore not suitable for publication in Nature Communications.

Response to reviewer and editorial comments

We thank the reviewers for their thoughtful comments. We regret any confusion arising from the inclusion of a “related manuscript”, a manuscript-in-preparation from our collaborators that shows application of Careless by a leading group in the field.

We agree with reviewer #1 that the “*general idea for this procedure seems fine*”, “*should be a fine approach*”, and “*the procedure seems fine in principle*”. The reviewer finds no fault with our approach itself beyond two contextual concerns—one about the application of Bayesian statistics and compatibility with other software, and the other about the performance of Careless relative to incumbent methods. Here, we address these concerns, as well as the editors’ concerns about whether our method can produce “clearly superior results”. We will also address reviewer #2’s comments on the “related manuscript” to the extent that they are relevant to the manuscript under consideration.

1. Is the approach novel?

The paradigm of Careless is a radical departure from the status quo. Algorithmically, it consists of a forward calculation of intensities based on a variational scale function and structure factor amplitudes. It is flexible and can accommodate explicit physical models with various error models, statistical priors, and scale functions (such as the multilayer perceptron described in the manuscript). The approach is feasible because of the use of variational inference, a statistical framework coming from physics which has recently seen a resurgence owing to its compatibility with concepts from deep learning. We are unaware of precedent for the use of statistical priors, deep learning scale functions, or variational inference to this end. In future work, we will describe extensions of this formalism, including the use of alternate, stochastic scale functions and multivariate priors that exploit the flexibility of this formalism. Indeed, the described implementation of Careless is the proof of concept which establishes a completely new paradigm.

Reviewer #1 observes that “*the procedure has the advantage that it is relatively general and does not require specification of the form of scale factors*”, but that “*existing methods also do not normally require such specification [of scale factors]*”. Although the reviewer does not provide examples, we concur that, for example, image scale factors and B-factors are allowed to vary as a function of image index, for example in HKL2000 and Aimless¹, and that coefficients of spherical harmonics can be fit that correct for uneven absorption of X-rays. The explicit mathematical expressions required and their multiplicative application restrict the range of corrections incumbent algorithms can perform. Careless enables the use of arbitrary scale functions. Indeed, it is well-known that multilayer perceptrons are universal function approximators, and hence the scale function in Careless can, in principle, take any form. Therefore, the sorts of corrections which Careless may learn are not restricted by the physical insight of a particular software engineer.

¹ Otwinowski & Minor, 1997; DOI 10.1016/S0076-6879(97)76066-X; Evans & Murshudov, 2013; DOI: 10.1107/S0907444913000061

2. Novel approaches contend with entrenched approaches

Reviewer #1 expresses concern about the performance of Careless relative to incumbent software (“... *although the procedure seems fine in principle, it is not clear that it actually works any better than existing methods.*”)

Our priority in writing this report was to explain the inference philosophy of *Careless* didactically and provide illustrative examples. For example, we refrained from using image layers in the first example, even though it can boost performance. Likewise we did not use a Student’s *t* distributed error model in the last example to keep the explanation tidy and sequential.

A key property of novel approaches, moreover, is that their performance leaves room for improvement, while incumbent software is usually close to optimal within the limitations of its paradigm. Across the three conventional software packages compared to (*Aimless*, *Precognition*, *cctbx.xfel*) each is only applicable to a specific domain of crystallography (monochromatic; polychromatic; and XFEL diffraction data, respectively) and has been optimized for years for that kind of data. Nevertheless, we reported performance competitive with *Aimless* and *CCTBX.XFEL* and outperform *Precognition*.

We believe that it is unreasonable to expect a single prototype to establish superiority against highly-optimized incumbent software in each domain of crystallography. Nevertheless, for the purposes of this response, we took some time to better illustrate its potential.

With respect to the first example (Sulfur-SAD diffraction), reviewer #1 comments that

“Fig. 3d does indicate that the density-modified map obtained with phenix autosol using data produced by Careless with 16 degrees of freedom is better than the map produced with data produced by Aimless. However the anomalous peak heights for the same data shown in Supp. Fig. 2 indicate in contrast that the data from Aimless are more accurate.”

As is clear from Figure 3c, *Careless* estimates anomalous differences better than *Aimless* at resolutions better than 3 Å. By contrast, *Careless* did worse at low resolution. This explains the seeming incongruence the reviewer observes: our maps look much better, because map appearance is most dependent on high-resolution (high spatial frequency) features. At the same time, anomalous peak heights are more sensitive to low-resolution reflections. We believe that this outcome is generally better than the reverse as it leads to more interpretable electron density. However, it is quite possible to obtain higher anomalous peaks with *Careless*.

To show this, we added an online example², added the results to Table S2 (referred to by the reviewer as “*Supp. Fig. 2*”), and added a reference to the example in the main text (“*As we illustrate in the online example “Boosting SAD signal with transfer learning”, it is possible to further improve the Careless output by using a simple transfer-learning procedure in which the parameters of the scale function are learned by a non-anomalous pre-processing step (see Table S2 for anomalous peak heights).*”; line 115). To keep the main text as accessible as possible, we chose to leave the simpler example in Figure 3.

² “Boosting SAD signal with transfer learning” at <https://github.com/Hekstra-Lab/careless-examples>

For convenience, we reproduce the new Table S2 here:

Site	Aimless	Careless (16 d.f.)	Careless (∞ d.f.)	Careless (image layers)	Careless (transfer)
Cys-6	14.91	11.97	10.45	14.48	15.86
Met-12	17.52	10.92	9.8	16.48	17.33
Cys-30	19.75	14.63	12.10	19.47	20.51
Cys-64	17.59	14.01	9.75	18.02	19.06
Cys-76	14.84	11.39	10.85	13.71	15.82
Cys-80	18.29	15.30	12.88	18.63	20.90
Cys-94	14.98	12.28	8.86	14.54	15.95
Met-105	19.89	14.84	11.67	20.30	21.48
Cys-115	19.38	16.13	11.41	18.92	20.29
Cys-127	14.62	10.64	10.16	14.50	15.35

The key changes in this example are:

1. Evaluation of the loss function depends on a sampling step. We increased the number of Monte Carlo samples used.
2. We applied image layers, which we introduce in the manuscript for the third example (see Figure 5).
3. We applied the idea of ‘transfer learning’, by using neural network weights from a non-anomalous preprocessing run.

The impact of 1 & 2 is recorded in the 4th column of the new Table S2. The 5th column demonstrates the gain from transfer learning from a preprocessing run. With these improvements, Careless robustly outperforms Aimless.

With respect to the third example (anomalous signal from XFEL data), reviewer #1 observed that “Similarly, Table 5f indicates that the Careless procedure is not better than the CCTBX procedure.”

Again, the Careless framework is flexible and extensible. In subsequent work, we will show that instead of using a univariate Wilson distribution as a prior, one can impose a bivariate prior. Using such priors can improve the detection of small differences between structures, as in anomalous differences or time-dependent conformational changes. We would like to reserve a (necessarily rather technical) description of this option for a future manuscript. However, to address the reviewer’s concerns, we here illustrate the power of Careless by applying a bivariate prior to the thermolysin data (Figure 5). Extending Table 5f with this analysis, we find that **Careless can dramatically outperform the incumbent software:**

Site	Careless	Careless (EO)	CCTBX	Careless (bivariate)
Zn-317	11.58	20.42	20.07	30.97
Ca-318	<3.5	5.42	6.17	8.48
Ca-319	<3.5	3.61	3.81	6.06
Ca-320	4.17	7.32	7.05	9.22
Ca-321	<3.5	5.49	7.25	7.99

[Extended Table 5f] Peak heights of anomalous scatterers in the thermolysin anomalous omit map, in σ units, for Careless output with and without Ewald offsets and conventional data processing with CCTBX. The rightmost column demonstrates strong gains in anomalous signal when using a multivariate prior to these XFEL anomalous diffraction data.

It is worth observing that the CCTBX software version we compare to represents a major recent advance in its own right. In Figure 5 of Brewster et al. (2019), the authors of the CCTBX software analyzed the same dataset to determine how many diffraction images are required to reach a given anomalous peak height for Zn-317. They show that for a 30-sigma peak height, they can reduce this number from 36,000 images to about 18,000 images, a two-fold gain in efficiency. With a bivariate prior, Careless achieves the same peak height with about 3,000 images, a further six-fold gain. To enable the reader to take advantage of this capability already, we have added an online example using this bivariate prior (“Using a bivariate prior to exploit correlations between Friedel mates”) and added a reference thereto on lines 242-243 of the main text.

3. Response to Comments by reviewer #2 on the “related manuscript”

We sought to illustrate the applicability of our software in the “real world” by submitting as a “related manuscript” a manuscript-in-preparation by our collaborators from the group of Dr. Richard Neutze—a leading group in the areas of time-resolved and serial X-ray crystallography. This was our first experience submitting a related manuscript and left some lessons to be learned. We would like to respond to the most pertinent comments by reviewer #2. The reviewer correctly observed that:

“In Results (Table 1): The overall values of CC1/2 and I/sigma given here are low and very low, respectively, and the overall value of CC1/2 has anyway little significance because it is biased by the falloff of intensity values with increasing resolution.”

“For the other two entries of the table, the following holds: CC is calculated from CC1/2 and thus carries no new information, and Rsplit is not given. Such a table should also detail high-resolution values for Rsplit, I/sigma, CC1/2 (or CC*) but these are not shown. Therefore the quality of the data is not only poor, but also remains poorly specified which is unexpected for a paper that claims superior data.”*

Regrettably, the reporting of crystallographic statistics in this manuscript-in-preparation were lacking. Most glaringly, the manuscript from the Neutze group reported the average $I/\sigma I$ for *unmerged* rather than for *merged* intensities (0.94 which would, indeed, be “very low” for merged intensities). The average $I/\sigma I$ for *merged* intensities from Careless is much higher ($\langle I/\sigma I \rangle = 20.7$ (overall), 1.72 (highest resolution shell; 2.9-2.85 Å)). In addition, Table 1 in the manuscript lacked highest-resolution shell statistics. Since we are not resubmitting the related manuscript, we reproduce the amended Table 1 below. It is now clear that the highest-resolution shell $CC_{1/2}$ for the Careless output (0.55) compares well to CrystFEL (0.34), even though the high-resolution limit has been extended from 3.4Å to 2.85Å. The same applies to R_{split} (37.8% versus 42% in the respective highest-res. shells). **That is, we extend the resolution of this dataset by >0.5 Å** (with the same or better high-resolution shell statistics).

	SRII:HtrII
Data collection	
Collection temperature (K)	293
Space group	P2 ₁ 22
Cell Dimension	
a, b, c (Å)	52.9, 67.7, 114.4
α, β, γ (°)	90, 90, 90
Resolution range	27.7 - 2.85 (2.9 - 2.85)
R_{split} (%) [1]	27.9 (37.8)
Mean $I/\sigma(I)$	20.7 (1.72)
$CC_{1/2}$	0.88 (0.55)
CC^* [2]	0.96 (0.84)
Completeness (%)	99.6 (99.9)
Reflections used in refinement	10,066 (997)
Reflections used for R-free	1,006 (100)
Number of unique reflections	10,094 (998)
Multiplicity	1068

Revised Table 1 for the “related manuscript”. We added values for R_{split} ; report mean($I/\sigma I$) for merged intensities, and added highest-resolution shell statistics. Refinement statistics are unaltered. Notes: [1] We calculated R_{split} , as described in the CrystFEL manual (compare_hkl: compare reflection data); that is, with optimization of a linear scale factor between half-datasets. [2] CC^* follows trivially from $CC_{1/2}$, as observed by reviewer #2.

4. Concerns about the use of Bayesian methods.

Reviewer #1 brings up general concerns about the application of Bayesian methods applied to X-ray data.

- a. *“Bayesian methods may give biased results: for example the French and Wilson calculation systematically overestimates values of weak amplitudes”.*

By way of background, French and Wilson sought to address the problem of observed negative X-ray diffraction intensities after correcting for background: although background scattering tends to vary smoothly across an image, the actual number of photons contributing to the total intensity of a diffraction spot varies stochastically. Since within a spot only the sum of ‘background’ and ‘signal’ photons is observed, one needs to use neighboring pixels to estimate the background contribution. This contribution can be overestimated due to counting noise or inaccurate foreground masks. In either case, for weak reflections the number of background-subtracted photons may be negative. Clearly, the true number of “signal photons” cannot be negative. French and Wilson addressed the question of how to correct for this by combining the observed intensity (e.g. -3 ± 5 photons observed) with a statistical prior on how many photons we should observe for a reflection (e.g., 5 ± 5 photons expected) to obtain the posterior distribution of the unknown true intensity (in this example, something like 2 ± 3).

Does this constitute a bias? Whether or not this change from -3 to $+2$ constitutes a *statistical bias* depends on whether our prior distribution fairly represents the statistical properties of the intensities (the prior is appropriate, so there is no statistical bias). Rather, French and Wilson introduced an *inductive bias* into their procedure—that structure factor amplitudes are positive. In our opinion, this is appropriate, just like in structure refinement it is appropriate to introduce the inductive bias that molecules consist of atoms with known patterns of covalent bonds. Such inductive biases, where appropriate, can greatly strengthen the power of inference methods. We further note that French-Wilson does not systematically provide larger posterior estimates for weak merged intensities *per se* (which would, indeed, be a problem), but rather for *noisy weak* observations. In other words, the reviewer’s concerns are understandable but unfounded.

Not all Bayesian methods are the same, of course. In particular, we apply a prior *during* scaling, not after-the-fact as is done in French-Wilson scaling. We contend that this is an improvement and, indeed, a significant innovation enabled by our formalism. Moreover, the modular, open-source nature of Careless implies that anyone who does not like the prior can readily incorporate their own favorite prior with minor modifications to the code (<https://github.com/Hekstra-Lab/careless/tree/main/careless/models/priors>).

- b. *“One concern with the Careless procedure is that the Bayesian posterior estimates that it produces are not well-suited for many of the downstream applications of the X-ray data”*

We understand (see below) that there are differences in the assumptions different downstream algorithms may make about how the data were treated. Careless provides the option to apply a

global B-factor to inferred structure factor amplitudes since a resolution-dependent decay in amplitudes is an expectation in some software (e.g. PHENIX's Xtrriage and Autosol).

Reviewer #1 refers to molecular replacement, and structure refinement as areas of concern. In our practice using Careless, we have not observed difficulties with molecular replacement or structure refinement, and do not see reason to discuss these topics at length. For example, we have done so successfully for each of our examples. For lysozyme, for example, we obtain a LLG > 5,000 during molecular replacement with Phaser in PHENIX, and $R_{\text{work}}/R_{\text{free}} = 15.2/17.2\%$ after simulated annealing and eight rounds of automated refinement in PHENIX using PDB ID 1VAT as a starting model (1VAT itself has $R_{\text{work}}/R_{\text{free}} = 20.5/23.2\%$ in the PDB). For the PYP, with basic refinement we likewise obtain $R_{\text{work}}/R_{\text{free}} = 20.8/22.3\%$, comparable with the original PYP statistics determined from a larger monochromatic dataset (PDB ID 2PHY, 18.6/22.6%). For thermolysin, refinement statistics are already reported in Figure 5. For the sensory rhodopsin data, molecular replacement and refinement also posed no difficulties, as described in the “related manuscript”.

It may be that challenges pop up with other proteins or downstream software applications—there are many possible combinations. We extended our final paragraph to read:

“In summary, we have described a general, extensible framework for inference of structure factor amplitudes from integrated X-ray diffraction intensities. We find the approach to be accurate, and applicable to a wide range of X-ray diffraction modalities. Careless is modular and open-source. We encourage users to report their experiences with downstream software, and to contribute extensions through <https://github.com/Hekstra-Lab/careless>. Careless provides a foundation for the ongoing development and systematic application of advanced probabilistic models to the analysis of ever more powerful diffraction experiments.”

REVIEWER COMMENTS

Reviewer #2 (Remarks to the Author):

First, I apologize that I reviewed the wrong manuscript. In retrospect, the reason was that there were, in the zip file for the reviewer, more files having to do with the "related manuscript" than for the submitted manuscript, and the abbreviations (like "rel_ms" and "art_file") which are part of the filenames may a posteriori be obvious to some people, but were not understood by me. (Nature Comm may want to consider improvements in the reviewer zip file layout to avoid such problems)

Concerning the manuscript "A UNIFYING BAYESIAN FRAMEWORK FOR MERGING X-RAY DIFFRACTION DATA": Following (in the lines between the ----- separators) are points that I noted before reading the comments of reviewer #1, and the rebuttal by the authors. Below, I use a star "*" to indicate where the rebuttal alters my conclusions.

The paper is quite original. It documents a novel statistical modeling and deep learning-based procedure for scaling crystallographic data, and extracting structure factor amplitudes as integral part of its application. The theoretical explanations appear to be sound and adequate, and are educational.

Three examples of different scaling scenarios are given, and the performance of the new method in these examples is compared to current methods. This is a weaker part of the paper, because when looking at the first example (sulfur SAD phasing) in depth, it turns out that the novel method is compared to a current method that performs poorly (compared to yet another current method), and the new method performs even worse (*). Since the assumption of the paper is that the current method (against which the comparison with the new method is made) represents the state-of-the-art, and this assumption is wrong, the resulting conclusions by the authors concerning the performance of the new method are questionable - if the new method would be as good as they claim, they should have obtained much better results.

My specific criticism concerning this example is:

- lines 92-95: The shadow shown in Figure 3a is close to the rotation axis. Since reflections on and near the rotation axis cannot be processed due to high Lorentz factor, this plays no role and no claim should be made that this leads to outliers. Furthermore, this is not at all a weak dataset since reflections are clearly visible to high resolution. However, it is true that the dataset is contaminated by the third harmonic, which leads to bad or missing data below 6Å resolution. The influence of bad or missing low-resolution data on anomalous peak heights is probably high.

The comparison of anomalous peak heights from CARELESS-scaling with those from AIMLESS-scaling for the HEWL data (Table S2 which likely shows "Anomalous difference Fourier map peak heights" rather than "Anomalous omit map peak heights") shows worse results for CARELESS than for AIMLESS (*). To obtain a different view on the data, this reviewer used a more common current method: the XDS program (4896 PDB entries released in 2021; DIALS-based deposition in 2021: 404) for processing of the deposited data, and SHELXC, SHELXD and

SHELXE for substructure solution and phasing. Program defaults were used throughout except that indexing used only reflections beyond 6Å resolution. Data processing and structure solution with these programs was trivial, and the calculations took less than 5 minutes of wallclock time on a \$1000 workstation. SHELXE reports the following anomalous peak heights:

::::

Anomalous density (in sigma units) at initial heavy atom sites

Site x y z occ*Z density

1	0.0668	0.6519	0.0170	16.0000	23.04
2	0.0096	0.6241	-0.1384	14.7472	22.22
3	-0.0697	0.8224	0.1989	14.7024	21.81
4	0.0194	0.6275	-0.1877	14.4256	23.06
5	0.1095	0.7328	-0.4187	14.0272	18.57
6	0.0792	0.7450	-0.1243	12.2064	18.98
7	0.0178	0.7984	0.2536	11.6912	16.40
8	-0.0902	0.8392	0.2030	11.4976	19.51
9	0.0081	0.7930	0.2067	11.2960	15.51
10	0.0893	0.7022	-0.4145	6.2688	5.59

::::

which are based SHELXE's phases from an unrefined and incomplete poly-Ala model of HEWL. The values in the "density" column above are higher than those in the AIMLESS column of table S2 (where "Mys-105" should be "Met-105"), which are in turn higher than those from CARELESS (16 d.f.). (The anomalous peak heights are also higher when using phases from a complete HEWL model with isotropic B factors, after rigid-body refinement.)

Comparing the CCanom of data from XDS with those from AIMLESS or careless (Fig 3c) shows significant differences; whereas XDS's CCanom is high (>80%) at low resolution and drops towards lower resolution, CCanom from AIMLESS and careless is low (at most 40 and ~58%, respectively) throughout the resolution range.

The processing and structure solution was repeated by me with frames 1-720, and was also successful. Taken together, this reveals that either the data processed by DIALS have a worse anomalous signal than those from XDS, or that both AIMLESS and careless treat the data in a suboptimal way such that the anomalous signal is not fully extracted. Any conclusions of the authors concerning careless's performance for accurate scaling of conventionally collected X-ray data should therefore be treated with caution, since they appear to be based on a comparison with one established way of processing and structure solution (DIALS/AIMLESS/PHENIX) which in this case performs particularly badly relative to another established way of processing and structure solution (XDS/SHELX*). In other words, what the authors describe as a difficult project is in reality not difficult.

- line 106: "In order to ensure a consistent origin, we supplied the sulfur atom substructure from the final refined model (PDBID: 7L84) during phasing." This gives more accurate sites than are available from Patterson or Direct methods, and is not available in a real-world

experiment where the structure is unknown at this point. More accurate sites result in a bias towards higher anomalous difference peaks.

- computing times and hardware should be given

Minor point:

- line 62: "the scale on which these observed intensities are expressed can vary ..." this is an uncommon usage of the word "scale", and the word "expressed" also does not fit - expressed by whom? The authors mean that the intensities can vary, due to different scale factors.

Supplement:

- line 68: "However, it implies that the structure factors output by Careless are on the same scale across resolution bins." I understand that this means that CARELESS produces E (normalized amplitude) values rather than F (structure factor amplitude) values, and I caution that application of a (Wilson) B-value to E values does not necessarily produce the F values that e.g. current refinement programs expect.

- The two lines of formulas after line 103 do not make sense to me

- line 140: The words "In French-Wilson scaling" are misleading. The French-Wilson algorithm is not about scaling; it is used to estimate the (positive) structure factor amplitudes, given the (potentially negative) measured intensities arising as a difference between peak counts and (extrapolated i.e. estimated) background counts.

After reading reviewer #1's comments and the authors' rebuttal, I note the following:

- the questions and criticisms of reviewer #1 are well-founded and balanced, and align with my own thoughts. I see no point in repeating them in my words, or commenting them and the authors' rebuttal, in this review

- the point that I marked with "*" above (i.e. the worse performance of CARELESS w.r.t. AIMLESS) is taken care of by the authors' rebuttal point "Boosting SAD signal with transfer learning". However my main point above remains - the comparison of a new method should be made against the best-performing current method.

- it is beyond my capabilities and time to analyze the two other examples in depth.

I do concede that comparisons with existing methods are tricky, and are prone to several types of biases. They should nevertheless be done, but the authors should probably tone down the claims and conclusions in this paper.

My summary is: the CARELESS framework appears to be an interesting addition to crystallographic methods, and future will tell which (and if) difficult crystallographic problems can be better resolved with it, and what can be learnt about applicability of deep-learning in crystallography using it as a starting point for further developments.

Reviewer #3 (Remarks to the Author):

The manuscript “A unifying Bayesian Framework for Merging X-ray Diffraction Data” by Dalton et al. presents a Bayesian solution for scaling and merging X-ray diffraction reflection intensities using a deep learning and variational inference approach, amendable to account for the potential influence of a variety of physical metadata.

Careless is first presented in the context of some, albeit simple, simulated data (a “toy” crystal), but then subsequently applied to three vastly different X-ray diffraction (experimental) dataset examples. The outcome is compared to results when substituting the scaling and merging analysis steps with incumbent software, specifically aimless, Precognition and cctbx.xfel.

The demonstration with simulated data, while simple, is a good illustration of the underlying principle of the algorithm and tractable for the ubiquitous reader. In general, the manuscript is an interesting read. Many more applications suffering from still poorly understood data scaling artifacts that could be explored with the application of Careless come to mind.

The only concern prevalent during the initial consideration of the manuscript (and shared with reviewer #1) is the risk of bias introduced by the Bayesian approach. However, the authors response in their rebuttal letter is fair; are they introducing a bias in their approach: to some extent, yes. Is this an unacceptable approach in the context of macromolecular crystallography? No. The authors do mention that this is indeed common practice, e.g. during structure refinement (not to mention molecular replacement), many “assumptions” on possible atom placement and covalent bond lengths are routinely made and considered “physically feasible” as opposed to “bias”.

Regarding the French and Wilson calculation, it is a long-standing debate within the community whether the implicit assumption that structure factors cannot (physically) have negative values should be made within software packages. Photon scattering is stochastic, and so are some detector responses. Therefore yes, negative intensities are observed after background subtraction. In my opinion, however, this is beyond the scope of the manuscript and reaching a community-wide consensus on this matter is not up to the authors. Since this does apply mainly to weak (high resolution) reflections, it would be of interest to include the resolution ranges for the Careless output compared to the incumbent software (as well as the merging statistics to gauge the quality and SNR of the data).

Each case is presented in a way that clearly illustrates the adaptability and modularity of the algorithm, which is the merit of careless and (I believe) the point the authors are trying to make. The case presentation is commendable, it does not oversell the “off the shelf” usability of careless by extensive (expert) parameter tuning nor does it omit the shortcomings and potential pitfalls for the inexperienced user. (As an aside, the results achieved with careless for the Laue experiment are indeed remarkable!)

With regard to reviewer #1’s concern that incumbent software outperforms careless in 2 out of 3 of the examples presented, this is indeed the case as presented. However this is not

detrimental to the manuscript nor the algorithm. Firstly, the point to highlight is the fact that what is being compared is one single software package dealing with 3 vastly different experimental set-ups and data manifestations on one hand, and 3 individual software packages with dedicated expertise on the other. Performance comparable to the incumbent software package dedicated towards analysing a particular type of data is therefore already a very good result.

Lastly, the reviewer's "concern with the Careless procedure [...] that the Bayesian posterior estimates that it produces are not well-suited for many of the downstream applications of the X-ray data" is true, but unfounded. An experienced macromolecular crystallography scientist will have encountered these discrepancies in both data format and definition of structure factor amplitudes between software packages before, particularly when dealing with difficult, non-standard cases. It is unlikely that careless will be the immediate choice for scaling and merging straight-forward macromolecular datasets in the future (simply due to the computational requirements compared to aimless), and it does provide the option of applying a global B-factor for conformity. This should not pose much of a hurdle for an experienced data analyst.

In summary, the software package careless implements a novel approach with great potential, and the modular and flexible design make it a strong tool for tackling difficult macromolecular crystallography datasets. The manuscript in general is well written, comprehensive and scientifically sound.

Dalton et al. – Response to reviewers

We thank the reviewers for their thorough review of our manuscript, “A Unifying Bayesian Framework for Merging X-ray Diffraction Data”. During review (spread over two cycles), the reviewers noted the originality of our work, and its potential to enable new X-ray diffraction experiments. They found the paper to be generally well written, comprehensive, and theoretically sound. The reviewers also agreed that Careless can outperform Aimless, our benchmark for the analysis of monochromatic rotation series data, using a “transfer learning” protocol we introduced in our initial revision. Implementations of this protocol, as well as a protocol that strongly boosts performance for our third example (serial X-ray Free Electron Laser data), are now available at <https://github.com/Hekstra-Lab/careless-examples> (henceforth referred to as ‘online’).

Here, we address the second round of reviewer comments (shown in *italics*). We will do so point-by-point. To assure that our response is complete, we reproduce all the reviewer comments from the second round of reviews.

A central concern by reviewer #2, was whether Aimless, our benchmark, represents the state of the art of monochromatic X-ray data processing. Per the editor’s request (by email, September 12, 2022), we therefore focus our response on the following two points,

- 1- Provide appropriate benchmarking of CARELESS against the best-performing methods, as highlighted by reviewer #2.
- 2- Demonstrate that CARELESS provides substantial advantages over existing software.

Reviewer #2 (Remarks to the Author):

Concerning the manuscript "A UNIFYING BAYESIAN FRAMEWORK FOR MERGING X-RAY DIFFRACTION DATA": Following (in the lines between the ----- separators) are points that I noted _before_ reading the comments of reviewer #1, and the rebuttal by the authors. Below, I use a star "" to indicate where the rebuttal alters my conclusions.*

The paper is quite original. It documents a novel statistical modeling and deep learning-based procedure for scaling crystallographic data, and extracting structure factor amplitudes as integral part of its application. The theoretical explanations appear to be sound and adequate, and are educational.

We thank the reviewer for this balanced assessment. Our examples were indeed chosen primarily for educational purposes.

Three examples of different scaling scenarios are given, and the performance of the new method in these examples is compared to current methods. This is a

weaker part of the paper, because when looking at the first example (sulfur SAD phasing) in depth, it turns out that the novel method is compared to a current method that performs poorly (compared to yet another current method), and the new method performs even worse ().*

For the benefit of the editor, we note that the reviewer's opinion was changed by our previous response. Our method performs well compared to Aimless, our chosen benchmark.

Since the assumption of the paper is that the current method (against which the comparison with the new method is made) represents the state-of-the-art, and this assumption is wrong, the resulting conclusions by the authors concerning the performance of the new method are questionable - if the new method would be as good as they claim, they should have obtained much better results.

The reviewer expanded on this comment in a further email correspondence on September 13:

to clarify my comments: from their presentation, it appears that the authors have done a good and thorough job in processing data of examples 2 & 3, and documenting their work. However, the same could have been said for example 1, and only by downloading freely available deposited raw data and re-processing with different programs it became clear that the comparison of their Careless processing (is the name of the program well chosen??) against another method is not against the best-performing method. As I tried to say, I don't have time to download and analyze in similar depth the data of their examples 2 and 3 so I cannot verify the authors' claim that Careless compares favourably or on par with existing best-performing methods. This brings (my) reviewing to its limits.

It would indeed be concerning if DIALS + Aimless were not a strong, *best-performing* benchmark. Below, we perform a thorough, point-by-point comparison of Careless, Aimless, and XDS, drawing on additional experiments and analysis, and an inquiry with the author of XDS. We find that Aimless compares favorably to XDS, with slight differences in performance, and fairly represents the state of the art. We have integrated several aspects of this further work into our manuscript (detailed below), as it led us to clarify crucial aspects of our analysis, more firmly establishes how Careless compares to incumbent software, and will be of interest to the many users of XDS.

My specific criticism concerning this example is:

- lines 92-95: The shadow shown in Figure 3a is close to the rotation axis. Since reflections on and near the rotation axis cannot be processed due to high Lorentz factor, this plays no role and no claim should be made that this leads to outliers. Furthermore, this is not at all a weak dataset since reflections are clearly visible to high resolution.

Shown below are the reflections provided to Careless (yellow circles) overlaid on the beam stop shadow as it appears in the first image of the data set. Several reflections do overlap the shadow region and are assigned nonzero Lorentz Polarization corrections by DIALS. Careless does not exclude any reflections, so in the context of our narrative the shadow region does contribute outliers. The number of outliers in this region is small, and thus the undulator harmonic is likely a much more significant source of error than this shadow. We adjusted the wording in the main text (lines 94-98) to indicate the relative significance of the two sources of outliers. To improve the flow of the text in light of these edits, we slightly rearranged the text on lines 94-101.

- However, it is true that the dataset is contaminated by the third harmonic, which leads to bad or missing data below 6Å resolution. The influence of bad or missing low-resolution data on anomalous peak heights is probably high.

Agreed.

The comparison of anomalous peak heights from CARELESS-scaling with those from AIMLESS-scaling for the HEWL data (Table S2 which likely shows “Anomalous difference Fourier map peak heights” rather than “Anomalous omit map peak heights”)

We apologize for this. The caption of Table S2 was correct (“Anomalous omit map peak heights from PHENIX refinement with isotropic B-factors and rigid body refinement.”), but reference to Table S2 in the main text (“The refined heavy atom structure factors yield similar sulfur peak

heights (Table S2)”) was, unfortunately, misleading and poorly separated from the preceding description of SAD phasing. Our lack of clarity may have led the reviewer to a different approach to inferring anomalous peak heights than ours.

We have split SAD phasing and anomalous peak height determination into separate paragraphs (at line 123), added a clarification of the procedure for anomalous peak height determination on lines 158-161 of the Supplementary Information under “Data collection and analysis”, and now restate lines 124-127 to read:

“Anomalous signal in real space, on cysteine and methionine S atoms, provides an additional measure of the accuracy of the estimated structure factor amplitudes. To this end, we performed limited automated refinement in Phenix using a sulfur-omit version of PDB ID 7L84 [19] as a starting model, and inferred peak heights from the resulting anomalous omit map.”

shows worse results for CARELESS than for AIMLESS ().*

For the benefit of the editor, we note this reviewer's opinion was changed by our previous response as denoted by the asterisk. Presumably this is because we showed superior performance to DIALS + Aimless using our transfer learning protocol. In addition, we now first state in the text that “As shown in Table S2, Aimless outperforms Careless with 16 d.f. and XDS in this regard, underscoring the subtle differences in the requirements each test imposes on the data.” (lines 127-128), which forms a suitable transition to the description of the “transfer learning” example in lines 129-134. We now restate the transfer learning results as follows (lines 131-135), which we hope fairly summarizes their relative performance:

“With this addition, Careless attains higher average anomalous peak height than Aimless and XDS (Table S2), better Spearman CCanom (Figure S6), and equal (by figure of merit) or better (by Bayes-CC and visual appearance) phased map quality (Table S1, Figure S7c). Relative performance may, of course, vary from dataset to dataset.”

To obtain a different view on the data, this reviewer used a more common current method: the XDS program (4896 PDB entries released in 2021; DIALS-based deposition in 2021: 404) for processing of the deposited data, and SHELXC, SHELXD and SHELXE for substructure solution and phasing. Program defaults were used throughout except that indexing used only reflections beyond 6Å resolution. Data processing and structure solution with these programs was trivial, and the calculations took less than 5 minutes of wallclock time on a \$1000 workstation. SHELXE reports the following anomalous peak heights:

.....

Anomalous density (in sigma units) at initial heavy atom sites

Site x y z occ*Z density
 1 0.0668 0.6519 0.0170 16.0000 23.04
 2 0.0096 0.6241 -0.1384 14.7472 22.22
 3 -0.0697 0.8224 0.1989 14.7024 21.81
 4 0.0194 0.6275 -0.1877 14.4256 23.06
 5 0.1095 0.7328 -0.4187 14.0272 18.57
 6 0.0792 0.7450 -0.1243 12.2064 18.98
 7 0.0178 0.7984 0.2536 11.6912 16.40
 8 -0.0902 0.8392 0.2030 11.4976 19.51
 9 0.0081 0.7930 0.2067 11.2960 15.51
 10 0.0893 0.7022 -0.4145 6.2688 5.59

which are based SHELXE's phases from an unrefined and incomplete poly-Ala model of HEWL. The values in the "density" column above are higher than those in the AIMLESS column of table S2 (where "Mys-105" should be "Met-105"), which are in turn higher than those from CARELESS (16 d.f.).

Firstly, we thank the reviewer for this very interesting analysis. We have done our best to repeat it and document the results in our manuscript. XDS is certainly a very popular tool, and a comparison increases the overall appeal of the manuscript.

Importantly, we note that the anomalous peak heights reported by SHELXE may be optimistic, particularly early in chain tracing. We include a description of this phenomenon at the end of this letter. Briefly, in our experiments we noted that initial SAD solutions typically have very high (~34 sigma) anomalous peak heights which decay as the phases improve and more of the chain is traced (see the final table in this letter). We suspect that this is the result of density modification being used to boost the anomalous signal early in model building by re-allocating density from the solvent region into the peaks. As more of the chain is traced, SHELXE may have less solvent region to work with and the anomalous peak heights decrease. Therefore, these numbers are not comparable to those from the anomalous omit maps reported in the manuscript. Furthermore, we note that the quality of initial solutions varies greatly between different inputs, and we are hesitant to report anomalous peaks from such protocols, and therefore do not currently describe these results in the manuscript. We included a brief explanation of this choice in a supplementary note (lines 162-165 of the SI).

(The anomalous peak heights are also higher when using phases from a complete HEWL model with isotropic B factors, after rigid-body refinement.)

We have added average peak heights from XDS to Table S2. The XDS peak heights are indeed higher than for Careless with the Student *t* error model presented in the main text. However, following our protocol, the XDS peak heights are neither higher than DIALS+Aimless nor Careless with transfer learning. We added a reference to this analysis on lines 127-128 of the main text. Further, we now show in Table S2 that combining XDS output with the Careless

transfer learning protocol increases the anomalous signal. That is, it appears that XDS output can be improved by re-scaling it in Careless. We added a description of this on lines 133-135 of the main text.

Comparing the CCanom of data from XDS with those from AIMLESS or careless (Fig 3c) shows significant differences; whereas XDS's CCanom is high (>80%) at low resolution and drops towards lower resolution, CCanom from AIMLESS and careless is low (at most 40 and ~58%, respectively) throughout the resolution range.

This is a very interesting point. It turns out that this is not a true gap in performance but a consequence of how CCanom is calculated in XDS. We reached out to the author of XDS, Dr. Wolfgang Kabsch, to understand how XDS reports anomalous correlation coefficients. We reproduce his explanation here verbatim:

The item 'Anomal Corr' printed in CORRECT.LP denotes the percentage of correlation between random half-sets of anomalous intensity differences. Correlations significant at the 0.1% level are marked by an asterisk.

It is computed the following way.

For each unique reflection all symmetry-related ones in the data set are collected together with their Friedel mates. Their intensities and variances are stored in a temporary array together with their +/- parity class.

Each +/- class is randomly splitted into halves of about equal size and an estimated mean intensity and its standard deviation is determined for each of the 4 halves; [I1+,sigma(I1+)], [I2+,sigma(I2+)], [I1-,sigma(I1-)], [I2-,sigma(I2-)]

4 anomalous differences are thus obtained

$$I11=(I1+)-(I1-)$$

$$I22=(I2+)-(I2-)$$

$$I12=(I1+)-(I2-)$$

$$I21=(I2+)-(I1-)$$

Weighted correlations $W*[I11,I22]$ and $W*[I12,I21]$ contribute to their corresponding resolution shell with weights

$$W=1/\{\text{var}(I1+) + \text{var}(I2+) + \text{var}((I1-) + \text{var}((I2-)+ 0.0025*[(I1+)^2 + (I2+)^2 + (I1-)^2 + (I2-)^2]\}$$

I have been using Diederichs & Karplus paper but have not updated the code along the more recent paper you mentioned in you e-mail. I have not paid too much attention to CCano because I

think other programs for structure determination like SHELX etc. will do a more detailed analysis anyway.

In other words, XDS reports weighted correlation coefficients with weights similar to, but not identical to, what one would use with standard maximum likelihood weighting and the assumption of normally-distributed, independent errors for each half-dataset. In addition to the inverse variance term for maximum likelihood, the XDS weights contain a hyperparameter, 0.0025, which downweights the influence of large intensities. Because this ad hoc term has not been extensively validated, we do not include correlation coefficients computed by this method in our manuscript.

To enable the reader to make a comparison with XDS output, however, we included three new figure panels in Figure S6. We summarize the impact of these new panels on lines 106-110, and specifically to the results for XDS on lines 119-120. The first panel uses the same Spearman correlation coefficient we use in Figure 3c to compare output from Aimless and XDS to the results from four scaling modes possible in Careless:

- (1) assuming a normally distributed error model, or
- (2) a Student *t*-distributed error model with 16 degrees of freedom,
- (3) the “transfer learning” protocol which we explained in our previous response to reviewers and refer to on lines 129-134 of the revised manuscript; and
- (4) use of image layers as introduced later in the manuscript.

By this comparison, XDS performs marginally better than Aimless. Careless provides superior CCanom over most of the resolution range for each modality, except when using a normal distribution for its error model, consistent with other observations in Figure 3.

In the third panel, we report inverse variance-weighted Pearson correlation coefficients which are equivalent to XDS with its 0.0025 hyperparameter set to zero. Evidently, this type of correlation coefficient is generally higher, not just for XDS, but also for the other scaling modalities. Indeed, XDS achieves an anomalous correlation coefficient of about 80% in the lowest resolution bin. By this measure XDS and Aimless perform nearly identically. In addition, they seem to slightly outperform Careless at low resolution, while Careless outperforms Aimless and XDS at intermediate and high resolution. We caution that these weighted correlation coefficients focus on just a subset of the data—the most accurate measurements. We consider the Spearman CCanom a more comprehensive and useful indicator. Nevertheless, we report both to facilitate comparison. For completeness, we also include the unweighted Pearson CCanom in the middle panel. Unweighted Pearson correlation coefficients are sensitive to noisy measurements and therefore not suitable for comparative analysis.

For completeness, we also reproduce here our implementation of Dr. Kabsch’s correlation coefficient. Because there is no published record of the approach, we would like to refrain from including it in the manuscript. As you can see, the differences with the new Figure

S6c are minimal and do not affect the conclusions.

The processing and structure solution was repeated by me with frames 1-720, and was also successful. Taken together, this reveals that either the data processed by DIALS have a worse anomalous signal than those from XDS, or that both AIMLESS and careless treat the data in a suboptimal way such that the anomalous signal is not fully extracted.

Any conclusions of the authors concerning careless's performance for accurate scaling of conventionally collected X-ray data should therefore be treated with caution, since they appear to be based on a comparison with one established way of processing and structure solution (DIALS/AIMLESS/PHENIX) which in this case performs particularly badly relative to another established way of processing and structure solution (XDS/SHELX). In other words, what the authors describe as a difficult project is in reality not difficult.*

In light of the new comparisons to XDS, we disagree with the assertion “*that either the data processed by DIALS have a worse anomalous signal than those from XDS, or that both AIMLESS and careless treat the data in a suboptimal way such that the anomalous signal is not fully extracted.*” As we describe, the initial anomalous difference peaks from SHELXE should not be taken to be a quantitative measure of the true anomalous signal. The anomalous peak height gap between XDS and Aimless disappears when a consistent phasing protocol is used (as shown in Table S2). The difference in CC_{anom} between the reviewer's XDS analysis and our Figure 3C are the result of a weighting scheme used by XDS, not a difference in performance.

Regarding the difficulty of this task, we have clarified our intent in the lines 97-101 wherein we state, “Conventional scaling and merging in Aimless or XDS [20 , 12] is successful for these data because these approaches use outlier rejection to explicitly identify and remove spurious reflections. These data therefore represent a challenging test case for our approach which considers all integrated reflections without outlier rejection.”

- line 106: *"In order to ensure a consistent origin, we supplied the sulfur atom substructure from the final refined model (PDBID: 7L84) during phasing." This gives more accurate sites than are available from Patterson or Direct methods, and is not available in a real-world experiment where the structure is unknown at this point. More accurate sites result in a bias towards higher anomalous difference peaks.*

To calculate experimental electron density maps for Figure 3d and new Figure S7, we supplied sulfur sites because the choice of origin is degenerate for this space group (No. 96). The origin is stochastically chosen by a heavy-atom search which complicates comparing maps. We do not report anomalous peak heights resulting from this analysis. In Table S2, we report calculated anomalous peak heights after automated refinement in Phenix against a S-omit atomic model of the protein, as described in the caption to the Table. We now further clarify this in the main text (lines 98-101) and in the SI (lines 158-165).

- *computing times and hardware should be given*

We have added the following note to the data availability section (see line 282):

"Each of these examples can be run on a consumer grade NVIDIA 3000 series GPU in under an hour."

Minor point:

- line 62: *"the scale on which these observed intensities are expressed can vary ..." this is an uncommon usage of the word "scale", and the word "expressed" also does not fit - expressed by whom? The authors mean that the intensities can vary, due to different scale factors.*

We restated this as:

"the observed intensities vary systematically due to physical artifacts correlated with the metadata, causing the reflections to be related to the squared structure factor amplitudes by different multiplicative scale factors, or scales" (line 64-65)

Supplement:

- line 68: *"However, it implies that the structure factors output by Careless are on the same scale across resolution bins." I understand that this means that CARELESS produces E (normalized amplitude) values rather than F (structure factor amplitude) values,*

Careless output is theoretically related to an E-value but not identical. To compute E values, Careless output should be divided by the square root of each reflection's multiplicity (compare, for example, the expression SI eq. 1 to section 7-10 of Rupp's textbook on Biomolecular Crystallography). Since this is a rather specialized issue, we chose to not alter the text.

and I caution that application of a (Wilson) B-value to E values does not necessarily produce the F values that e.g. current refinement programs expect.

So far we have had few issues with downstream software. However, we anticipate user feedback will help identify issues and lead to mitigation strategies. We were advised by a PHENIX developer that enforcing an estimated Wilson B-factor during inference would potentially increase compatibility with their tools. Therefore, we provide this as an option to users. We added a sentence highlighting this option in the SI (lines 70-71).

- The two lines of formulas after line 103 do not make sense to me

We have edited this section (lines 103-106 of the SI) to be more clear. The major point of confusion was probably that the function f_{θ} returns a two-vector.

- line 140: The words "In French-Wilson scaling" are misleading. The French-Wilson algorithm is not about scaling; it is used to estimate the (positive) structure factor amplitudes, given the (potentially negative) measured intensities arising as a difference between peak counts and (extrapolated i.e. estimated) background counts.

We agree. On lines 39 and 41, we now refer to this as "French-Wilson corrections".

After reading reviewer #1's comments and the authors' rebuttal, I note the following:

- the questions and criticisms of reviewer #1 are well-founded and balanced, and align with my own thoughts. I see no point in repeating them in my words, or commenting them and the authors' rebuttal, in this review

- the point that I marked with "" above (i.e. the worse performance of CARELESS w.r.t. AIMLESS) is taken care of by the authors' rebuttal point "Boosting SAD signal with transfer learning". However my main point above remains - the comparison of a new method should be made against the best-performing current method.*

We have added comparisons to XDS in Tables S1 and S2 and Figures S6 and S7, and demonstrated synergistic processing with XDS and Careless, as described above. We trust this will satisfy the reviewer's concerns that we do not draw comparison to state-of-the-art software.

- it is beyond my capabilities and time to analyze the two other examples in depth.

I do concede that comparisons with existing methods are tricky, and are prone to several types of biases. They should nevertheless be done, but the authors should probably tone down the claims and conclusions in this paper.

We are willing to make specific changes to the text of the manuscript where that still seems appropriate in light of our discussion of the reviewer's initial concerns.

My summary is: the CARELESS framework appears to be an interesting addition to crystallographic methods, and future will tell which (and if) difficult crystallographic problems can be better resolved with it, and what can be learnt about applicability of deep-learning in crystallography using it as a starting point for further developments.

We agree with this. We believe our contribution represents a milestone in the application of deep-learning methods to crystallographic data and is, indeed, 'a starting point for further developments.' that will transform which 'difficult crystallographic problems' can be addressed.

Reviewer #3 (Remarks to the Author):

The manuscript "A unifying Bayesian Framework for Merging X-ray Diffraction Data" by Dalton et al. presents a Bayesian solution for scaling and merging X-ray diffraction reflection intensities using a deep learning and variational inference approach, amendable to account for the potential influence of a variety of physical metadata.

Careless is first presented in the context of some, albeit simple, simulated data (a "toy" crystal), but then subsequently applied to three vastly different X-ray diffraction (experimental) dataset examples. The outcome is compared to results when substituting the scaling and merging analysis steps with incumbent software, specifically aimless, Precognition and cctbx.xfel.

The demonstration with simulated data, while simple, is a good illustration of the underlying principle of the algorithm and tractable for the ubiquitous reader. In general, the manuscript is an interesting read. Many more applications suffering from still poorly understood data scaling artifacts that could be explored with the application of Careless come to mind.

We appreciate these thoughtful comments. We too are excited to test the limits of experimental artifacts which may be corrected by our method.

The only concern prevalent during the initial consideration of the manuscript (and shared with reviewer #1) is the risk of bias introduced by the Bayesian approach. However, the authors response in their rebuttal letter is fair; are they introducing a bias in their approach: to some extent, yes. Is this an unacceptable approach in the context of macromolecular crystallography? No. The authors do mention that this is indeed common practice, e.g. during structure refinement (not to mention molecular replacement), many "assumptions" on possible atom placement and covalent bond lengths are routinely made and considered "physically feasible" as opposed to "bias". Regarding the French and Wilson calculation, it is a long-standing debate within the community whether the implicit assumption that structure factors cannot (physically)

have negative values should be made within software packages. Photon scattering is stochastic, and so are some detector responses. Therefore yes, negative intensities are observed after background subtraction. In my opinion, however, this is beyond the scope of the manuscript and reaching a community-wide consensus on this matter is not up to the authors. Since this does apply mainly to weak (high resolution) reflections, it would be of interest to include the resolution ranges for the Careless output compared to the incumbent software (as well as the merging statistics to gauge the quality and SNR of the data).

We note that in contrast to the work presented in the related manuscript from the Neutze lab, we did not extend the resolution of the data in our manuscript. Indeed, we decided to set the same resolution limit for Careless as for the benchmark software to facilitate comparison. To make it easier to assess data quality, we have now included some additional statistics for each data set. These can be found in new supplementary tables S3, S4, and S5.

Each case is presented in a way that clearly illustrates the adaptability and modularity of the algorithm, which is the merit of careless and (I believe) the point the authors are trying to make. The case presentation is commendable, it does not oversell the “off the shelf” usability of careless by extensive (expert) parameter tuning nor does it omit the shortcomings and potential pitfalls for the inexperienced user.

This was indeed what we intended to convey in our presentation. As observed in the first round of reviews, the drawback of this presentation was that we did not immediately bring out all of the ‘bells and whistles’ to the first example. By referencing the additional transfer learning example on lines 129-134, inclusion of additional examples online, and inclusion of additional benchmark results in Figures S6 and S7 and Tables S1 and S2, we sought to achieve a balance between showcasing Careless’ performance relative to incumbent software, while retaining the didactic flow of the manuscript.

(As an aside, the results achieved with careless for the Laue experiment are indeed remarkable!)

For anyone familiar with polychromatic X-ray diffraction experiments, it is clear that lack of open-source, modern software has paralyzed this field for the past 25 years. The incumbent software, Precognition, lacks a primary publication describing its algorithms, its source code is closed and cannot be modified, and basic data processing statistics, like CC1/2, have not been implemented, and its scaling routine can not handle datasets of more than a few hundred diffraction images. Careless finally provides a free, performant, and open-source alternative. As an aside, we are collaborating with DIALS developers to develop an open-source alternative to Precognition for the remaining data reduction steps from indexing through integration.

With regard to reviewer #1's concern that incumbent software outperforms Careless in 2 out of 3 of the examples presented, this is indeed the case as presented. However this is not detrimental to the manuscript nor the algorithm. Firstly, the point to highlight is the fact that what is being compared is one single software package dealing with 3 vastly different experimental set-ups and data manifestations on one hand, and 3 individual software packages with dedicated expertise on the other. Performance comparable to the incumbent software package dedicated towards analysing a particular type of data is therefore already a very good result.

We agree with the spirit of this comment—no other software has been shown to be applicable across the crystallography modalities that Careless can handle (monochromatic rotation data, polychromatic data, and time-resolved/serial crystallography stills). Moreover, we have established that Careless can outperform both Aimless and XDS on recovery of anomalous signal when using a transfer learning approach, now provided as an online example.

We, moreover, established that Careless can outperform state-of-the-art software suites for the processing of serial X-ray crystallography data, usually collected at X-ray free electron laser (XFEL) facilities. Specifically, there are two main software suites in use today: cctbx.xfel and CrystFEL. In the revised manuscript, we show that Careless can perform on par with cctbx.xfel using its state of the art error model with a protocol recommended by the authors¹. In our first response, we further showed that with upcoming enhancements, Careless surpasses the performance of cctbx.xfel, corresponding to a large decrease in the number of diffraction images required to reach a desired level of anomalous signal. We now provide an example of how to do so online, and refer to this example on lines 258-260.

The “related manuscript” provided in our initial submission (which may not have been available to reviewer #3), further showed that Careless can improve on CrystFEL for serial crystallography data. In particular, Careless extended the useful resolution range of serial Sensory Rhodopsin data by 0.5 Å. This manuscript awaits submission by our collaborators.

In summary, in each case state-of-the-art, or superior, results are attainable with Careless.

Lastly, the reviewer's “concern with the Careless procedure [...] that the Bayesian posterior estimates that it produces are not well-suited for many of the downstream applications of the X-ray data” is true, but unfounded. An experienced macromolecular crystallography scientist will have encountered these discrepancies in both data format and definition of structure factor amplitudes between software packages before, particularly when dealing with difficult, non-standard cases. It is unlikely that Careless will be the immediate choice for scaling and merging straight-forward macromolecular datasets in the future (simply due to the computational requirements compared to Aimless), and it does provide the option of applying a global B-factor for conformity. This should not pose much of a hurdle for an experienced data analyst.

This has indeed been our experience so far.

¹Brewster AS, Bhowmick A, Bolotovskiy R, Mendez D, Zwart PH, Sauter NK. SAD phasing of XFEL data depends critically on the error model. Acta Cryst D. 2019 Nov 1;75(11):959–68.

In summary, the software package careless implements a novel approach with great potential, and the modular and flexible design make it a strong tool for tackling difficult macromolecular crystallography datasets. The manuscript in general is well written, comprehensive and scientifically sound.

We thank the reviewer for the positive feedback, and we agree with their assessment that Careless represents a novel approach with great potential.

Walk-through of reviewer #2's analysis

Prompted by reviewer #2, we re-processed out lysozyme sulfur SAD data using XDS for integration, scaling, and merging followed by SHELX for SAD phasing. For XDS, we followed the protocol as recommended by the reviewer. We ran SHELXC/D/E from the xdsGUI with default parameters for SHELXC, the following parameters for SHELXD:

Find 10 heavy atoms of type 5 Use data from 999.0 to 2.3 A resolution
Allow sites on special position? Yes No
Limit number of tries to 1000
Exclude E-values less than 1.50
 Resolve 4 disulphide bridges

(all default except for specifying the heavy atom and disulfides)

and for SHELXE:

Phase structure based on given sites
Invert heavy atom substructure for phasing? try both enantiomorphs
Native data do include heavy atoms
Use fractional solvent content of 0.45
Side chain tracing: LOAD
 Modify data defaults
 Extend diffraction data to 1.75 A.
 High resolution cut-off for input data 0.10 A.
 Highest resolution for calculating phases from .pda 1.80 A.
 Autotracing
Cycles 3
 Keep model 1 cycles autotracing
 Beta Anti-parallel
 Helical Standard
 Non-crystallographic symmetry
Order 2

With this protocol, we find that SHELXE reports:

Anomalous density (in sigma units) at initial heavy atom sites

Site	x	y	z	occ*Z	density
1	0.6245	0.9879	0.0951	16.0000	27.00
2	0.7964	0.9849	0.4839	11.7120	17.17
3	0.6519	0.9349	0.2724	9.8912	15.96
4	0.8233	1.0656	0.4630	9.2352	16.02
5	0.7222	0.8797	-0.1405	9.1568	14.08
6	0.7371	0.8896	-0.1790	8.4112	11.54
7	0.6453	1.0047	-0.3511	7.0640	9.78
8	0.6806	0.9404	0.4806	5.4880	8.43
9	0.6987	0.9175	-0.1819	5.3120	8.43
10	0.7025	0.7572	-0.3511	5.1712	7.77
11	0.7465	0.9208	0.1395	4.6208	10.86
12	0.6391	0.8953	0.2468	3.8848	8.24
13	0.6281	0.9602	-0.0522	3.2384	6.41

While the peak heights appear to be a remarkable result, we find it is not indicative of useful anomalous signal. Increasing the number of Autotracing cycles to 100 and re-running SHELXE, we found that SHELXE only built 59 of 129 residues, and it could not reliably differentiate the two possible enantiomorphs for this space group. The final figure of merit was 0.241 which is

below the cutoff of 0.25 typically considered the minimum and by no means a guarantee of SAD phasing. The chain trace looked discontinuous and unrealistic as well.

Although, we concede there is a certain amount of subjectivity to this point.

We concluded from this test that the heavy atom substructure determined by SHELXD was likely not accurate enough to consider this a proper solution.

We decided to make a more thorough test by ensuring that SHELXD and SHELXE were both run to convergence. We increased the number of heavy atom search tries to 10,000 in SHELXD and the number of Autotracing cycles in SHELXE to 100. In this case we should have a better initial heavy atom substructure and a more complete model and therefore better phase estimates. In this test we saw SHELXE was able to trace nearly the full chain, 112 residues of a total 129. It could clearly differentiate the two possible enantiomorphs for this space group. The final figure of merit was an excellent 0.655. The final model had only 2 chains instead of 4 and a clearly globular fold.

Surprisingly, from this more converged analysis, we obtained the following anomalous peak report:

Site	x	y	z	occ*Z	density
1	0.6239	0.9890	0.1029	16.0000	20.15
2	0.7964	0.9842	0.4846	13.0768	12.24
3	0.6503	0.9348	0.2684	11.3792	18.06
4	0.8234	1.0700	0.4578	11.0848	16.04
5	0.7199	0.8775	-0.1438	10.9200	12.48
6	0.7382	0.8882	-0.1741	8.0096	12.06
7	0.7452	0.9205	0.1283	7.2032	13.94
8	0.6449	1.0026	-0.3465	7.0272	4.33
9	0.9350	1.0600	0.6427	6.0704	2.47
10	0.7006	0.7586	-0.3500	4.6048	4.35
11	0.7542	1.0214	-0.1439	4.1232	4.82

Which is much lower yet more inline with the peak heights we have recorded in our manuscript.

Therefore, we find that XDS and SHELXC/D/E can do an excellent job processing these data. However, the anomalous peaks reported by SHELXE must be interpreted with care. Apparently, in the case of an inaccurate, incomplete, or partly incorrect heavy atom substructure, these metrics can be inflated substantially. We are not experts in SHELX, but we suspect that this may be due to density modification routines which seek to increase the density at anomalous sites during initial phasing by flattening solvent regions. Early in structure solution it can be difficult to accurately identify solvent regions as new electron density continually appears while the phases are improving. It can be the case that improving the phases by tracing more of the chain actually decreases rather than increases the anomalous peaks. We suspect this has to do with less of the unit cell being classified as solvent for density modification as more of the model is built. We reach the paradoxical conclusion that a better initial solution may have lower anomalous peak heights after chain tracing.

Heavy atom tries	1,000	1,000	1,000	10,000	10,000	10,000
Autotracing cycles	0	3	100	0	3	100
Figure of merit	0.301	0.251	0.241	0.347	0.386	0.655
Residues built	—	27	59	—	51	112
No. chains	—	4	4	—	4	2
Highest peak	34.13	27.00	29.07	35.09	23.09	20.15

SHELXD/E Results for various numbers of heavy atom placement attempts and autotracing cycles.

Based on the anomalous peak heights determined from our PHENIX refinement protocol and from a well-converged SHELXE run, XDS and SHELX do not perform significantly better than DIALS integration paired with Aimless or Careless (transfer). Indeed, DIALS + Careless (transfer) + AutoSol results in a final model with 127 out of 129 residues built in a single chain, a model-map CC of 0.86, and $R_{\text{work}}/R_{\text{free}}$ of the Autobuilt model of 20.5 / 22.9% without any further

processing). Of course, these are popular tools and their performance may differ depending on the context.

Reviewers' Comments:

Reviewer #2:

Remarks to the Author:

I am referring to the latest cycle of revision, and the "Response to reviewers".
I think that the authors have done enough to warrant publication of the article.